# INTERCEPT CANCER: CANCER PRE-SCREENING WITH LARGE SCALE HEALTHCARE FOUNDATION MODELS

## ABSTRACT

Cancer screening, leading to early detection, saves lives. Unfortunately, existing screening techniques require expensive and intrusive medical procedures, not globally available, resulting in too many lost would-be-saved lives. We present CATCH-FM, CATch Cancer early with Healthcare Foundation Models, a cancer pre-screening methodology that identifies high-risk patients for further screening solely based on their historical medical records. With millions of electronic healthcare records (EHR), we establish the scaling law of EHR foundation models pretrained on medical code sequences, pretrain compute-optimal foundation models of up to 2.4 billion parameters, and finetune them on clinician-curated cancer risk prediction cohorts. In our retrospective evaluation comprising of thirty thousand patients, CATCH-FM achieves strong efficacy, with 50% sensitivity in predicting first cancer risks at 99% specificity cutoff, and outperforming feature-based tree models and both general and medical LLMs by up to 20% AUPRC. Despite significant demographic, healthcare system, and EHR coding differences, CATCH-FM achieves state-of-the-art pancreatic cancer risk prediction on the EHRSHOT few-shot leaderboard, outperforming EHR foundation models pretrained using on-site patient data. Our analysis demonstrates the robustness of CATCH-FM in various patient distributions, the benefits of operating in the ICD code space, and its ability to capture non-trivial cancer risk factors. Our code will be open-sourced.

## 1 INTRODUCTION

Early cancer detection by cancer screening is one of the most effective ways to combat cancer (Siegel et al., 2024). Cancers detected at an early stage are treated with significantly improved patient outcomes (Haue et al., 2024; Kim et al., 2024). Recent medical advancements also significantly improved the curative rates for cancers detected at early stages (Ju et al., 2024; Maru & Jaffee, 2024; Springfeld et al., 2023; Thiele et al., 2024). Routine screening with follow-up monitoring of high cancer risk patients is standard practice, enabling timely intervention and effective treatments (Altmayer et al., 2024; Gyawali & Booth, 2024; Rubinstein et al., 2024).

Despite its benefits, cancer screening remains underutilized (Zhang et al., 2022), especially in populations with limited healthcare resources (Xu et al., 2024b) due to reliance on invasive, resource-intensive procedures like medical imaging (Rohatgi et al., 2020; Washington & Deville, 2020). Screening is more common for cancers with clear risk factors, like breast and colorectal (Siegel et al., 2025), while cancers without early symptoms, such as pancreatic cancer, often progress silently and are detected late, with survival durations under one year (Blackford et al., 2024).

We present CATCH-FM: CATch Cancer early with Healthcare Foundation Models, a cancer pre-screening tool that identifies high-risk patients using only their medical history. CATCH-FM is pretrained on large-scale longitudinal EHR data and finetuned on clinician-curated cancer cohorts. It directly operates on precise medical codes (ICD), learning general medical patterns through next-code prediction and finetuned to capture cancer risk signals reflected in patient medical history (Lee et al., 2021; 2022; Phan et al., 2020). Once trained, CATCH-FM can be deployed in EHR systems to predict cancer risk at low cost, supporting healthcare providers in deciding when and whom to screen.

To facilitate the study of EHR foundation models in cancer pre-screening, we build NHIRD-Cancer, a cancer risk prediction benchmark, by sampling more than three million patients from the Taiwanese

National Health Insurance Research Database (NHIRD) (Hsieh et al., 2019), a government de-identified and research-accessible EHR database. We allocate three million patients for pretraining, consisting of billions of medical events spanning two decades, and the remainder for finetuning and evaluation. We focus on three cancers based on their critical needs for early detection: pancreatic, liver, and lung cancers (Kim et al., 2024; Kukhareva et al., 2024; Thiele et al., 2024), and curate clinically reliable cancer cohorts after matching cancer patients with control-to-case groups.

With billions of medical events available, we examine how compute budget (FLOPs), model size, and pretraining tokens affect performance. Our findings establish an EHR foundation model scaling law and confirm the benefit of large-scale pretraining on EHR data. Accordingly, we pretrain compute-optimal CATCH-FM models up to 2.4b parameters. When finetuned and evaluated for cancer risk prediction on NHIRD-Cancer, CATCH-FM consistently outperforms feature-based tree models and language models trained on the same data. Without loss of flexibility, it demonstrates strong predictive efficacy, achieving over 50% and 70% sensitivity in the *first* and *subsequent* target cancer cohorts at a 99% specificity cutoff, and reaching 50% and 80% AUPRC, respectively, offering strong reassurance when ruling out cancer (Grimes & Schulz, 2005).

On the pancreatic cancer risk prediction task from Stanford Medicine EHRSHOT benchmark (Wornow et al., 2023), CATCH-FM outperforms the prior state-of-the-art EHR foundation model, CLMBR (Wornow et al., 2023), pretrained on millions of on-site patient records, while CATCH-FM is trained on data from drastically different populations, disease prevalence, and coding systems (ICD vs. SNOMED). Our analyses further demonstrate the robustness of CATCH-FM across different patient cohorts, preexisting conditions, and pre-screening configurations. Our interpretability analyses following the method of Gao et al. (2024) reveal that CATCH-FM identified not only known cancer risk factors but also non-trivial markers discovered in recent medical research.

We view CATCH-FM as an effective, low-risk, and widely performable pre-screening tool that can assist healthcare providers make informed, effective, and efficient cancer screening decisions. To facilitate future research and development, our data curation and modeling code will be open-sourced under MIT license. The NHIRD-Cancer benchmark and trained model checkpoints will be released under the same license as NHIRD, enabling reproducibility and future research within the necessary constraints of privacy and regulations on patient data.

## 2 RELATED WORK

Cancer screening significantly improves prognosis and patient outcomes (Kim et al., 2024; Kukhareva et al., 2024; Thiele et al., 2024). Advancements in cancer treatment have made cancers more treatable and potentially curable, if they are detected in early stages (Chu et al., 2024; Hu et al., 2024; Kim et al., 2024; Liu et al., 2024). Recent advances in AI-powered medical imaging, like CT scans with multimodal models, have enhanced cancer screening accuracy, sometimes exceeding human-level sensitivity (Cao et al., 2023; Chen et al., 2023; Wang et al., 2024b; Xu et al., 2024a). The challenge is that medical imaging is resource-intensive, inaccessible for under-served populations, and not widely performable (Elmohr et al., 2024; Truhn et al., 2024; Vrudhula et al., 2024; Waite et al., 2021).

Using electronic health records (EHR) to assess cancer risk is a promising path to improve cancer screening effectiveness and efficiency, i.e., by identifying patients with high cancer risk for healthcare professionals to make informed cancer screening decisions (Lee et al., 2021; 2022). Recent approaches have made attempts using feature-based machine learning models on large, task-specific datasets to detect cancer risks, and have shown the possibility of capturing cancer risk signals in EHR (Peduzzi et al., 2024; Placido et al., 2023b).

Large language models pretrained on medical corpora are effective on text-oriented medical tasks (Clusmann et al., 2023) such as medical question answering (Singhal et al., 2025), clinical document summarization (Liu et al., 2025), and radiology report generation (Sun et al., 2024b). Recent research has also pretrained foundation models on structured EHR data (Choi et al., 2016; Gao et al., 2020; Yao et al., 2019), and explored their utilities in various clinical tasks (Savcisens et al., 2024; Sun et al., 2024a; Yang et al., 2023b). A challenge for EHR foundation models is that many EHR datasets only cover a slice of patients' healthcare records (Faltys et al., 2021; Johnson et al., 2016; Pollard et al., 2018), e.g., MIMIC-IV mainly focuses on emergency departments and ICU encounters (Johnson et al., 2016). Foundation models pretrained on EHR slices are more effective

Table 1: Statistics of our NHIRD subset.

| Item | Count |
|---|---|
| # of patients | 3,989,369 |
| # of visits (In/out-patients, pharmacy) | 1,441,453,071 |
| # of diagnosis (ICD9 codes) | 1,491,556,480 |
| # of procedures (Surgeries) | 46,282,990 |
| # of treatments (Non drug treatments) | 1,729,212,760 |
| # of medications (Drug prescriptions) | 2,770,631,039 |
| # of unique medical codes | 185,138 |
| # of tokens for all codes | 7,923,387,479 |
| # of patients for pretraining (80%) | 3,191,495 |
| # of patients for finetuning (20%) | 797,874 |
| Avg. # of visits per patient | 271 |

Table 2: Statistics of *first* and *subsequent* target cancer screening benchmark.

| | Pancreatic | Liver | Lung |
|---|---|---|---|
| *First* Target Cancer Cohort | | | |
| Total | 285,097 | 266,563 | 277,943 |
| Positive | 4,520 | 5,092 | 3,985 |
| Negative | 280,577 | 261,471 | 273,958 |
| Pos./Neg. Ratio | 1.61% | 1.95% | 1.45% |
| *Subsequent* Target Cancer Cohort | | | |
| Total | 277,381 | 265,066 | 263,662 |
| Positive | 7,381 | 5,835 | 4,648 |
| Negative | 270,000 | 259,231 | 259,014 |
| Pos./Neg. Ratio | 2.73% | 2.25% | 1.79% |

in corresponding prediction tasks such as emergency department triage (Sun et al., 2024a) and ICU readmission risk (Jiang et al., 2023).

Many tasks, like cancer risk prediction, require longitudinal EHR data to capture full patient history. Foresight uses two decades of data to pretrain Transformers for biomedical forecasting (Kraljevic et al., 2022). MOTOR trains a 143M-parameter model on time-to-event tasks using 2.7M patient records from 2014–2022, effectively predicting diagnosis time (Steinberg et al., 2023). CLMBR, a 141M-parameter Transformer, is pretrained on 2.57M Stanford Medicine records over three decades (Wornow et al., 2023), with 6.7K records released in the EHRSHOT benchmark for 15 few-shot prediction tasks (Wornow et al., 2023).

Due to privacy and ethical constraints, large-scale longitudinal EHR data are hard to release publicly. Most EHR foundation models are pretrained on one or two hospital sites and are limited in scale to around 100M parameters (Steinberg et al., 2023; Wornow et al., 2023). Their advantage over feature-based models in cancer risk prediction remains unclear, for instance, while CLMBR performs well on procedure outcomes and lab predictions, it fails to outperform feature-based models in pancreatic cancer prediction on EHRSHOT (Wornow et al., 2023).

## 3  NHIRD-CANCER BENCHMARK

Initially, we overview our source data, NHIRD, and then the curation process of NHIRD-Cancer.

### 3.1  NHIRD PRELIMINARY

**Overview.** The National Health Insurance Research Database (NHIRD) include electronic health record (EHR) of over 99.99% of Taiwan population. It includes decades of de-identified records from all healthcare encounters under the National Health Insurance program, with diagnoses, prescriptions, and procedures (Hsieh et al., 2019). Coding standards are applied uniformly across providers, ensuring consistency and correctness. A sample patient record is shown in Appendix Figure 6b.

**Availability.** The sensitivity of longitudinal EHR data makes large-scale public access challenging (Guo et al., 2023). Among existing longitudinal EHRs (Kraljevic et al., 2022; Steinberg et al., 2023; Wornow et al., 2023), NHIRD is among the most accessible for open research. Eligible institutions in Taiwan and their international collaborators can access NHIRD with IRB approval or exemption under standardized protocols. Access is regulated by the Ministry of Health and Welfare to ensure strong privacy protections (Hsieh et al., 2019; Lin et al., 2018; Sung et al., 2020b). See Appendices A and B for details on de-identification, data quality, and access to NHIRD.

### 3.2  NHIRD-CANCER BENCHMARK CURATION

Our cancer screening benchmark is curated collaboratively with clinicians, following their guidance in the selection of positive group and cohort control-case group.

**Data Statistics.** We accessed a subset of NHIRD via collaboration with a Taiwanese medical school, with IRB exemption, as the study uses anonymized, non-human-subjects data. The dataset includes 3+ million patients from 1996–2013, covering 1.4 billion visits with diagnoses, treatments, and medications coded in ICD-9 (CDC). It provides 15+ years of physician-confirmed, government-validated history and over 8 billion medical codes. Summary statistics are in Table 1, with preprocessing and

additional details in Appendix A. Our subset aligns with full NHIRD averages (18 vs. 16.5 visits per patient over 15 years) and is comparable in scale to the 3.67M-patient records from Stanford medicine used in CLMBR (Wornow et al., 2023), which is not publicly available.

**Targeted Cancers.** We focus on pancreatic, liver, and lung cancers, which have high mortality rates and increased risk of metastatic cancer (De Visser & Joyce, 2023; Gupta & Massagué, 2006; Ji et al., 2023) where early detection offers substantial benefit (Kukhareva et al., 2024; Thiele et al., 2024; Blackford et al., 2024; Haue et al., 2024). The screening task is a binary classification: predicting whether a patient will later develop the target cancer. As suggested by clinicians, we focus on predicting target cancer *one year* after a clinical visit, enabling early intervention for proactive care. We set the available medical history per patient to be five years, following clinical standards. We explicitly split the benchmark into *first* target cancer (patients with no prior cancer history) and *subsequent* target cancer (patients with a history of other cancers but no prior target cancer) cohorts, and conduct experiments on models' effacacy on both scenarios.

**Case and Control Group.** Guided by clinicians, we rigorously follow the standard case-control study methodology and previous cancer screening studies on NHIRD (Phan et al., 2020; Lee et al., 2021; 2022) to create case-control dataset. We use the first occurance of target cancer type as the positive case. The (negative) control group includes patients with no cancer diagnoses, ensuring a clear distinction between cases and controls, and are matched with the case group as follows.

1. Patient Demographic Matching: Control patients are matched to case patients based on age and gender, to minimize confounding factors from demographics.
2. Relative Duration Matching: Control patients are then filtered to have clinical visits on the same index date as the case patient's diagnosis date, ensuring aligned medical timelines. They are also matched with a comparable length of medical history up to the index date.
3. Cumulative Duration Matching: Finally, controls are matched to case patients with the same total duration of lifetime medical history.

Our case-control matching goes beyond demographics by incorporating history length and timing, aligning patients more closely so differences reflect cancer status and enhancing validity.

**NHIRD-Forward.** We further collect an external dataset, referred to as NHIRD-Forward, from a different sample of the healthcare system at a distinct timeline from 2016-2021, 3 years after our NHIRD samples. The hospital only provided *first* liver and lung caner cohorts for testing. It follows the same NHIRD format while representing a different patient cohort with no overlap to NHIRD-Cancer, and is exclusively used to evaluate zero-shot generalization.

**Dataset Processing.** We use three-digit ICD-9 codes *157*, *155* and *162* to define pancreatic, liver, and lung cancers, respectively. Control patients must not have cancer-related codes from *140–239* (top level of Neoplasms). To support broader cancer types and maximize early detection (Chen et al., 2020), we use top-level ICD codes to capture more cases. Table 2 lists benchmark statistics. All cancer diagnoses are confirmed by licensed physicians and validated by the government. More details in Appendix A.

## 4 METHODS

This section presents the model, training, and inference of CATCH-FM.

### 4.1 MODEL

Figure 1 illustrates the model setup of CATCH-FM. It represents the healthcare record of each patient as a medical code sequence and models them by a decoder-only transformer.

**Patient Representation.** In longitudinal structured EHR databases, the patient records are often represented by a coding system, for example, ICD (CDC) or SNOMED (Cornet & de Keizer, 2008), using a unique ID to encode each specific piece of information. A typical patient record can start with their demographic information, such as age $c_{\text{age}}$ and gender $c_{\text{gender}}$, followed by a sequence of $n$ chronologically ordered visits, $v_1, v_2, ..., v_n$. Each visit $v_i$ consists of $m$ medical events, $c_i^1, c_i^2, ..., c_i^m$, covering diagnoses, medications, procedures, and other clinical events.

Figure 1: An example input sequence for a patient record and the architecture of CATCH-FM.

Specifically, CATCH-FM represents each patient's EHR token sequence $x$ as,

$$[c_{\text{age}}, c_{\text{gender}}, v_1, t_1, v_2, \ldots, v_{n-1}, t_{n-1}, v_n, \texttt{[EOS]}]; \qquad v_n = [c_n^1, c_n^2, \ldots, c_n^m]. \qquad (1)$$

The demographic tokens age $c_{\text{age}}$ is discretized into predefined categorical ranges and gender $c_{\text{gender}}$ is assigned as a distinct token. Each medical event code $c$ is assigned a unique token. Time intervals between consecutive visits are captured using time tokens $t$, which are also discretized into predefined categories to encode temporal information. A special token, $\texttt{[EOS]}$ marks the end of the record.

Though each ID can be mapped into the language space, for example, to their names and descriptions, CATCH-FM directly operates in the ID space which is more compact—each event is encoded by one token—and precise without ambiguity.

**Architecture.** CATCH-FM uses the standard decoder-only transformer architecture $G_{\theta}$ on top of the patient's EHR token sequence $x$. We use rotary positional embeddings (RoPE) (Su et al., 2023) to encode positional information for visits:

$$\text{RoPE}(\boldsymbol{x_i}, p_i) = \boldsymbol{h_i} \cdot \cos(\theta(p_i)) + \boldsymbol{h_{i\perp}} \cdot \sin(\theta(p_i)). \qquad (2)$$

It assigns the same absolute position $p_i$ (i.e., replacing sequential positions like 0,1,2,3... with 0,0,1,1...) to event tokens $\boldsymbol{x_i}$ from the same visit. This allows the model to capture the relationships between events within and across visits.

## 4.2 Pretraining, Finetuning, and Inference

CATCH-FM employs the standard pretraining, finetuning, and inference pipeline.

**Pretraining.** We pretrain CATCH-FM $G_{\theta}$ *from scratch* on our healthcare record pretraining dataset, using the standard autoregressive next-token prediction objective:

$$\min_{\theta} \mathcal{L}_{\text{LM}}, \quad \mathcal{L}_{\text{LM}} = -\frac{1}{n} \sum_{j=1}^{n} \log f_{\theta}(x_j | \boldsymbol{x}_{<j}), \qquad f_{\theta}(x_j | \boldsymbol{x}_{<j}) = \text{Softmax}(\boldsymbol{E} \boldsymbol{h}_j). \qquad (3)$$

The next code probability $f_{\theta}(\cdot)$ is computed on token embeddings $\boldsymbol{E}$ and hidden states $\boldsymbol{h}$ from $G_{\theta}$. Pretraining on large scale patient record with autoregressive language modeling task enables CATCH-FM to capture the complex medical patterns in patient health trajectories, for example, associations of medical events and potential risk factors of diseases.

**Finetuning.** For the cancer prediction task, we employs supervised fine-tuning on the pretrained CATCH-FM $G_{\theta}$. It uses cross-entropy loss to learn whether a patient will be diagnosed as a target cancer outcome $y$.

$$\min_{\theta, \phi} \mathcal{L}_{\text{SFT}}, \quad \mathcal{L}_{\text{SFT}} = -\log f_{\phi}(y \mid \boldsymbol{x}_{\texttt{[EOS]}}), \quad f_{\phi}(y \mid \boldsymbol{x}_{\texttt{[EOS]}}) = \text{Softmax}(\boldsymbol{W} \boldsymbol{h}_{\texttt{[EOS]}} + \boldsymbol{b}). \qquad (4)$$

The learnable parameters include $\theta$ in the foundation model, and $\boldsymbol{W}$ and $\boldsymbol{b}$ from a linear prediction layer. Finetuning makes CATCH-FM specialized for cancer risk prediction by capturing risk factors from medical histories while leveraging general medical knowledge learned from pretraining.

**Inference.** The inference of CATCH-FM is to take a patient's medical history from their EHR record, run a forward pass of the pretrained and finetuned CATCH-FM instances, and predict the cancer risk of the patient. The inference is efficient as only one prediction token is needed per patient.

The sole requirement of patient history makes CATCH-FM a nature fit for cancer prescreening. Healthcare providers can deploy CATCH-FM on a large amount of patient EHR data. The predicted high-risk patients can then be further evaluated by professionals to determine who and when to undergo cancer screening. An accurate prescreening thus effectively triages patients, improving the efficiency and efficacy of cancer screening. It can potentially increase the cancer screening rate by providing decision evidences for healthcare professionals and raise awareness from patients.

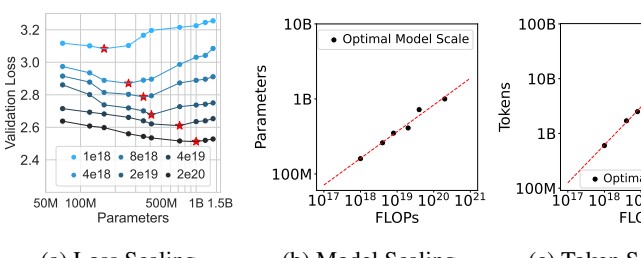

Table 3: Compute-optimal CATCH-FM instances pre-trained with variant FLOPs, and their estimated GPU hours in typical A100 machines.

(a) Loss Scaling  (b) Model Scaling  (c) Token Scaling

Figure 2: Scaling Law of CATCH-FM, including loss IsoFLOPs, estimated FLOP-optimal parameters and pretraining tokens.

| FLOPs | Parameters | A100 Hours |
|-------|-----------|-----------|
| 1e18 | 160m | ∼16 |
| 4e18 | 260m | ∼64 |
| 2e19 | 410m | ∼128 |
| 4e19 | 720m | ∼256 |
| 2e20 | 1b | ∼1280 |

## 5 EXPERIMENTAL METHODOLOGIES

**Dataset.** We use the NHIRD-Cancer benchmark described in Sec. 3.2 for the pretraining, finetuning, and in-domain evaluation. We randomly partition the dataset, allocating 80% for pretraining and 20% for cancer screening benchmarks, ensuring no overlap between them. Each benchmark is further split into 80%/10%/10% for train/valid/test, using stratified sampling to maintain consistent class distributions for *first* and *subsequent* target cancer cohorts.

To evaluate model generalization, We use NHIRD-Forward (Sec. 3.2) to assess zero-shot generalization where all patient cohort are used for evaluation. In addition, we use the EHRSHOT benchmark (Wornow et al., 2023), containing 6,739 Stanford Medicine patients (1990–2023), primarily coded in SNOMED, CPT, and RxNorm. It introduces significant distribution shifts from NHIRD: different healthcare systems, different medical codings, and different populations with different cancer prevalence. We use EHRSHOT's official pancreatic cancer risk prediction task following their few-shot setting (1–128 positives). It is the only available cancer risk prediction task. We transfer EHRSHOT patients to NHIRD by exact code matching when possible and cosine-similarity embedding soft matching (threshold 0.98) for unmapped codes. Appendix C details the process.

**Evaluation metrics.** NHIRD-Cancer evaluations use AUROC, AUPRC, specificity, and sensitivity, with the first two serving as the main metrics. AUROC measures the model's ability to discriminate between patients with and without cancer across all decision thresholds. AUPRC captures the precision–recall trade-off in imbalanced tasks in cancer risk prediction. Specificity, the ratio of true negatives to all negatives, reflects reliability in identifying low-risk patients. Sensitivity, the ratio of true positives to all positives, reflects how well the model flags cancer-risk patients. When possible, we evaluate sensitivity at 99% specificity, a common (pre)screening threshold (Cao et al., 2023; Halner et al., 2023; Jopek et al., 2025), and a clinician-defined standard to reduce false positives, ensuring clinical trust.

**Baselines.** We include standard tree-based models with bag-of-words as input (Bharati et al., 2023): XGBoost (Chen & Guestrin, 2016) and LightGBM (Ke et al., 2017), and well-established EHR Deep learning models (Wang et al., 2024a): StageNet (Gao et al., 2020) and RETAIN (Choi et al., 2016). We compare with pretrained language models, by converting medical codes into text and finetuning BioGPT (Luo et al., 2022) and Qwen2.5 (Yang et al., 2024) on the same data. For EHRSHOT, we compare CATCH-FM against methods reported in their leaderboard (Wornow et al., 2023).

**Implementation Details.** We implement CATCH-FM using a standard architecture (Biderman et al., 2023), a decoder-only transformer with rotary position embeddings and flash attention, and pretrain from scratch on NHIRD. Medical codes, demographics, and special tokens form a 185,138 token vocabulary, with sequences capped at 2,048 tokens. Longer histories are chunked without overlap during pretraining and truncated for fine-tuning. More architecture, hyperparameters, and training details are provided in Appendix D.

## 6 EVALUATION RESULTS

This section presents experimental results evaluating CATCH-FM's scaling law, effectiveness, generalization, and ablation studies, as well as analyses of its captured cancer risk factors.

Table 4: Downstream scaling behavior of compute-optimal CATCH-FM on *first* and *subsequent* cancer cohorts. The *first / subsequent* positive rates are shown in parentheses. Sensitivity is evaluated at a decision threshold corresponding to a fixed 99.0% specificity cutoff. All results are averages over five random seeds. The highest value for each metric is **bold**.

| Model | Pancreatic *first / subsequent* (1.59% / 2.73%) | | | Liver *first / subsequent* (1.95% / 2.25%) | | | Lung *first / subsequent* (1.45% / 1.80%) | | |
|---|---|---|---|---|---|---|---|---|---|
| | AUROC | AUPRC | Sensitivity | AUROC | AUPRC | Sensitivity | AUROC | AUPRC | Sensitivity |
| XGBoost | 91.6 / 95.4 | 26.3 / 68.4 | 31.0 / 61.9 | 91.2 / 95.8 | 36.2 / 69.7 | 36.3 / 65.2 | 91.4 / 95.3 | 27.6 / 69.4 | 32.3 / 66.5 |
| LightGBM | 91.5 / 95.8 | 25.6 / 69.1 | 31.9 / 62.1 | 92.0 / 95.9 | 35.4 / 69.9 | 36.2 / 66.9 | 91.5 / 95.6 | 24.3 / 69.2 | 31.1 / 69.3 |
| RETAIN | 68.9 / 21.6 | 3.4 / 2.0 | 0.0 / 0.0 | 74.8 / 20.3 | 5.2 / 1.4 | 0.0 / 0.0 | 74.2 / 82.6 | 3.9 / 8.9 | 0.0 / 0.0 |
| StageNet | 64.3 / 69.7 | 2.4 / 4.5 | 0.0 / 0.0 | 59.9 / 64.1 | 2.4 / 3.1 | 0.0 / 0.0 | 66.9 / 64.1 | 2.3 / 3.1 | 0.0 / 0.0 |
| BioGPT-347m | 91.8 / 93.7 | 19.5 / 50.1 | 19.9 / 42.0 | 88.6 / 93.3 | 24.9 / 49.1 | 22.4 / 42.2 | 89.5 / 90.7 | 19.6 / 50.0 | 24.9 / 48.6 |
| Qwen2.5-500m | 90.3 / 92.7 | 22.3 / 57.9 | 25.4 / 50.8 | 90.4 / 93.7 | 32.4 / 60.3 | 32.4 / 55.9 | 86.3 / 92.8 | 15.9 / 60.1 | 18.8 / 53.1 |
| CATCH-FM-160m | 91.4 / 97.1 | 42.4 / 79.6 | 43.1 / 75.3 | 89.3 / 96.1 | 39.3 / 76.6 | 39.1 / 73.6 | 89.2 / 94.1 | 33.0 / 74.8 | 38.9 / 73.1 |
| CATCH-FM-1b | 93.5 / 97.2 | 57.2 / 82.9 | 58.6 / 78.6 | 91.3 / 96.3 | 49.6 / 76.7 | 48.9 / 74.3 | 91.1 / 95.7 | 47.5 / 77.7 | 52.6 / 75.3 |
| CATCH-FM-2.4b | **94.4 / 97.8** | **61.3 / 84.7** | **60.6 / 80.8** | **92.2 / 96.6** | **52.8 / 79.0** | **53.6 / 75.8** | **92.6 / 96.3** | **49.6 / 80.2** | **53.1 / 79.6** |

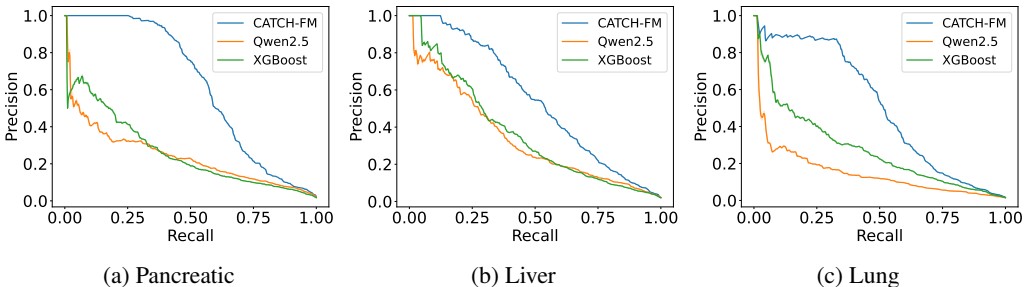

| (a) Pancreatic | (b) Liver | (c) Lung |
|---|---|---|

Figure 3: AUPRC comparison across cancer types with CATCH-FM-2.4b, Qwen2.5-500M, and XGBoost evaluated on the *first* target cancer cohorts.

## 6.1 SCALING LAWS OF PRETRAINING ON HEALTHCARE RECORDS

To pretrain effectively on healthcare records, we first conduct a thorough scaling law analysis of CATCH-FM. Specifically, we use IsoFLOP profiling (Hoffmann et al., 2022): pretraining models with various sizes at target FLOPs by varying the number of pretraining tokens.

Figure 2a illustrates clear parabola-shaped IsoFLOP curves, with different model sizes achieving minimum validation loss at various FLOPs. Using these data points, we fit a power law to characterize the relationship between FLOPs ($C$), the loss-optimal model size ($N_{\text{opt}}$), and the optimal number of training tokens ($D_{\text{opt}}$), as illustrated in Figures 2b and 2c:

$$\text{Optimal Model Sizes: } N_{\text{opt}} \propto C^{0.34}, \qquad \text{Optimal Token Counts: } D_{\text{opt}} \propto C^{0.69}. \qquad (5)$$

Scaling laws in healthcare foundation models resemble the scaling laws of large language models (Hoffmann et al., 2022), highlighting the potential of large-scale EHR foundation models. While emergent capabilities need further study, we pretrain and finetune a series of compute-optimal CATCH-FM models across various FLOPs (Table 3) and evaluate their efficacy in cancer risk prediction. The benefits of compute-optimal pretraining are detailed in Appendix F.

## 6.2 OVERALL RESULTS

Table 4 shows the overall performance of cancer risk prediction in NHIRD-Cancer. CATCH-FM outperforms XGBoost and LightGBM, strong tree models that often outperformed EHR foundation models (Wornow et al., 2023), by 20%+ AUPRC on *first* target cancer cohorts and 15%+ AUPRC on *subsequent* target cancer cohorts. It also significantly outperforms medical (BioGPT) and general (Qwen) large language models pretrained on the texts of NHIRD. Sect. 6.4 further studies the benefits of pretraining directly on medical codes rather than converting codes into natural language. Moreover, our studies show that CATCH-FM achieve 50% and 70% sensitivity in *first* and *subsequent* cohorts at a decision threshold corresponding to 99% specificity, confirming its ability to identify high-

Table 5: Zero-shot evaluation on NHIRD-Forward dataset for *first* liver and lung cancer.

| Methods | AUROC | AUPRC | Sensitivity |
|---------|-------|-------|-------------|
| **Cancer**: Liver, **Pos./Neg. (Ratio)**: 5798/297459 (1.95%) | | | |
| XGBoost | 93.3 | 41.8 | 40.7 |
| Qwen2.5-500m | 91.9 | 30.0 | 29.1 |
| CATCH-FM-2.4b | **93.6** | **44.8** | **42.6** |
| **Cancer**: Lung, **Pos./Neg. (Ratio)**: 5024/347974 (1.44%) | | | |
| XGBoost | 93.7 | 34.7 | 37.9 |
| Qwen2.5-500m | 89.1 | 13.6 | 17.2 |
| CATCH-FM-2.4b | **94.1** | **36.4** | **41.4** |

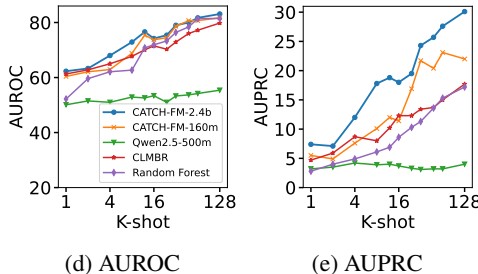

(d) AUROC          (e) AUPRC

Figure 4: AUROC and AUPRC on EHRSHOT pancreatic cancer from their public leaderboard.

risk patients for further screening (high sensitivity) and avoid unnecessary patient distress (high specificity). Additional details are found in Appendix E.

Cancer risk prediction is a challenging task, particularly for *first* target cancer cohorts, where many baselines fail to achieve meaningful performance. CATCH-FM demonstrates strong effectiveness, achieving 50%+ AUPRC in predicting *first* cancer cases. For *subsequent* target cancer cohorts, CATCH-FM effectively leverages prior cancer diagnoses as strong risk indicators and achieves 80%+ AUPRC. The benefit of scale is evident in all cohorts. CATCH-FM-2.4b outperforms CATCH-FM-160m by 10–15% AUPRC on *first* cancer cohorts and by 5–10% on *subsequent* cohorts. Scaled-up foundation models perform better on the harder *first* cancer prediction task.

## 6.3 GENERALIZATION ABILITY

We evaluate CATCH-FM under two generalization settings: temporal shifts within the same healthcare system (NHIRD), and distributional shifts across different systems and countries (EHRSHOT).

**Across Hospital Site and Time.** We evaluate CATCH-FM on cohorts from an adjacent timeline within the same hospital system, targeting *first* liver and lung cancer. Table 5 shows zero-shot results on NHIRD-Forward, demonstrating robustness to temporal shifts in population. Additional robustness analyses on cohort variations (G) and history exclusion windows (H) are in the Appendix.

**Across Healthcare System and Country.** Figure 4 shows the K-shot results of CATCH-FM on the EHRSHOT official pancreatic cancer leaderboard, using K positive and K negative on-site examples for tuning. Despite significant shifts in population, healthcare systems, and more than 50% medical code mismatch, CATCH-FM achieves state-of-the-art across all shots on the EHRSHOT leaderboard with a only handful of on-site examples, maintaining the best AUROC and AUPRC.

Moreover, CATCH-FM-160m outperforms CLMBR (140m) in AUROC for K ≥ 8 and in AUPRC, with only minor gaps at a lower shots, while scaling to CATCH-FM-2.4b further boosts few-shot performance, highlighting the benefits of scale. Notably, CLMBR is pretrained at a similar scale with on-site data (2.57M Stanford patients, same as EHRSHOT), while CATCH-FM faces distribution shifts with only 43% of SNOMED codes mapped to ICD. As ICD is already the predominant U.S. EHR standard (Feinstein et al., 2023), such SNOMED-to-ICD issues occur in only a fraction of healthcare providers. Overall, these results highlight CATCH-FM's robustness to distribution shifts and code mapping loss, underscoring its potential for transfer across healthcare systems and sites.

## 6.4 ABLATION STUDY

Table 6 shows ablation studies on different components of CATCH-FM. Modeling patient history is a key source of evidence for CATCH-FM. Demographic information alone is a poor indicator; pretraining is a key advantage of CATCH-FM; different model architectures, similar to observations in large language models, yield mild differences.

Converting the medical codes into their corresponding textual names and pretraining CATCH-FM on texts hinders performance. As shown in Figure 4a, the loss quickly drops to near zero, a strong sign of overfitting. The model memorizes code names and predicts them trivially after seeing the initial tokens. Figure 4b shows the loss is only meaningful on the first token, with the rest memorized. How to better adapt general domain LLMs to EHR data is an interesting future research direction.

Table 6: Ablation study on various pretraining strategies and model architecture on cancer pancreatic task at 160m with *first* cancer cohorts.

| Variations | AUROC | AUPRC | Sensitivity |
|---|---|---|---|
| **Medical Code Representation** | | | |
| CATCH-FM | **91.4** | **42.4** | **43.1** |
| demographic-only features | 50 | 1.8 | 0.0 |
| No Pretrain | 86.9 | 16.7 | 18.7 |
| **Model Architecture** | | | |
| w. Token Level Rel-Pos | 90.8 | 42.0 | 42.1 |
| w/o. time token | 90.6 | 41.2 | 41.3 |
| **Converted Text Representation** | | | |
| demographic-only features | 50 | 2.0 | 0.0 |
| No Pretrain | 90.5 | 18.8 | 17.5 |
| Finetune Pythia | 89.9 | 16.7 | 18.8 |
| Pretrain from Scratch | 86.5 | 14.5 | 15.5 |
| Continual Pretrain Pythia | **91.4** | 25.8 | 29.4 |

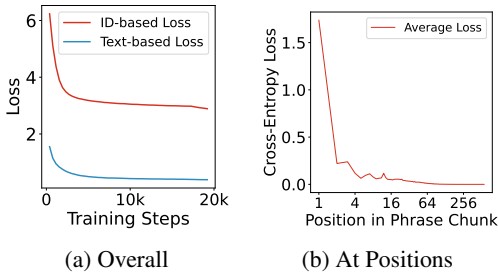

(a) Overall  (b) At Positions

Figure 5: Pretraining losses of CATCH-FM (a) on medical code IDs and texts and (b) at relative text token position in ID name phrase chunk.

Table 7: Top latent features on *first* cancer patients from CATCH-FM's prediction layer.

| Cancer | Features Learned by Sparse Autoencoder (Gao et al., 2024) with Description Generated by Node2Graph (Foote et al., 2023) |
|---|---|
| **Pancreatic** | High blood pressure, Diseases of the respiratory system, Operations on larynx or trachea, Telmisartan |
| | Type 2 diabetes mellitus, Hypertensive heart disease, Norvasc tablet, Hypothyroidism and thyroid |
| | Infectious and parasitic diseases, Intestinal infectious diseases, Pramipexole, Piracetam |
| **Liver** | Duodenal ulcer, Tetracyclines, Ursodeoxycholic acid, Famotidine |
| | Repair of uterus and supporting structures, Diagnosis on lymphatic structures, Ancillin, Ketoprofen |
| | Diclofenac, Anxiety state, Dysthymic disorder, Gastrojejunal ulcer |
| **Lung** | Loperamide, Chest view, Chronic ischemic heart disease, Diovan, Lymphatic diagnostic procedures |
| | Hypertensive disease with congestive heart failure, Isosorbide mononitrate, Inguinal hernia repair |
| | Esophagomyotomy, Norvasc tablet, Isosorbide mononitrate |

## 6.5 RISK FACTORS CAPTURED

This experiment leverages recently developed LLM interpretation method (Gao et al., 2024; Kang et al., 2025) to understand the risk factors captured by CATCH-FM. We train a sparse autoencoder on the prediction layers of finetuned CATCH-FM on *first* target cancer patients and then leverage neuron-to-Graph (N2G) (Foote et al., 2023) to explain the top active latent features on positive cancer cases. Additional implementation details are found in Appendix I.

Table 7 identifies features corresponding to top cancer risk factors captured by CATCH-FM. It includes not only trivial factors, such as Type 2 diabetes (Cui & Andersen, 2012) and high blood pressure (Stocks et al., 2012), but also non-trivial ones such as hypothyroidism, thyroid disorders, and Norvasc, with some indicators only presented at top medical journals less than ten years ago (Sarosiek et al., 2016; Wang et al., 2018b), after our training data cutoff.

## 7 CONCLUSION

CATCH-FM provides a new cancer risk prediction methodology by pretraining and finetuning Transformers on millions of patients' medical code sequences. Its effectiveness (50%+ and 70%+ sensitivity on *first* and *subsequent* target cancer cohorts), low risk (99% specificity), and wide applicability (only requiring inference on EHR records) make it a natural fit for cancer pre-screening. It helps healthcare professionals efficiently decide whom and when to screen for cancer, potentially improving the effectiveness and coverage of cancer screening and ultimately, patient outcomes. Our experiments on NHIRD and EHRSHOT demonstrated the benefit of scale in pretraining EHR foundation models, their generalization ability across significantly different healthcare systems, and their ability to capture non-trivial cancer risk factors. We hope our findings, analyses, and open-source codes can inspire and facilitate further research and deployments in leveraging AI to solve real-world healthcare problems.

ETHICS STATEMENT

Our research was conducted using de-identified EHRs obtained through formal research collaboration under regulated access and IRB review, with a determination of exemption made as the study relied solely on secondary, anonymized data and involved no human subjects. We ensured strict compliance with data governance rules set by the dataset sources for access. Privacy and confidentiality were preserved at all times, and no attempt was made to re-identify individuals. We restrict the release of model weights to authorized researchers with the same data access approvals, thereby safeguarding against misuse. Our work aims to advance early cancer detection through trustworthy artificial intelligence and foundation models while acknowledging potential biases in historical health data and the need for careful validation before any clinical deployment.

REPRODUCIBILITY STATEMENT

To ensure the reproducibility of our paper, a complete description of the NHIRD data schema, preprocessing pipeline, and full dataset statistics is detailed in the appendix. For transparency, we will publicly release our preprocessing, modeling, and training code under the MIT license, along with detailed instructions for reproducing the experiments. While direct redistribution of NHIRD data is restricted, all eligible institutions can apply for access under the same IRB framework. We will release pre-trained model checkpoints to authorized researchers with equivalent NHIRD data access for replication.

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

## A NHIRD Details

The Taiwanese National Health Insurance Research Database (NHIRD) is one of the largest and most comprehensive de-identified population-wide EHR datasets globally (Hsieh et al., 2019), covering over 99% of the Taiwanese population. It contains longitudinal medical records spanning over two decades, including patient demographics, diagnoses, prescriptions, clinical events, medical procedures, and hospital visits from all hospitals and medical facilities in Taiwan. Maintained by the National Health Insurance Administration in collaboration with the Ministry of Health and Welfare, the NHIRD comprises both registration files and hospital claim data submitted for reimbursement under the National Health Insurance (NHI) program.

We leverage a subset, which includes a randomly selected cohort of three million patients from 1996 to 2013, of the Taiwanese National Health Insurance Research Database (NHIRD) for pretraining EHR foundation models and constructing clinical downstream tasks for benchmarks. Unlike other datasets, the NHIRD stands out for its scale and comprehensiveness under a single-payer healthcare system, which enables standardized data collection and creates a comprehensive and lifelong record of patients' medical footprints. This makes NHIRD one of the most suitable resources for the real-world implementation of EHR foundation models. The NHIRD contains three main categories of medical information, with all personal details, such as ID, birthdate, and residential postcode, de-identified:

- **Demographics:** This includes details about medical institutions (e.g., centers, hospitals, and clinics) and de-identified data on patients, physicians, and pharmacists.

- **Visit:** Comprehensive records of medical visits, including outpatient clinic visits (including visit to medical centers and hospitals), hospitalizations, and pharmacy drug fulfillments.

- **Order details:** Detailed records of prescriptions, procedures, medical equipment, and materials associated with each type of visit.

Table 8 lists all NHIRD tables with descriptions. The three main visit tables are CD (outpatient clinic visits, including medical centers and hospitals), DD (hospitalization), and GD (pharmacy), each linked to an order table: OO (for CD), DO (for DD), and GO (for GD). For CATCH-FM, we use ID, CD, OO, DD, DO, GD, and GO tables, as they constitute complete patient medical histories. Demographic and other statistics to further describe data density and sparsity are shown in Table 9 and Figure 7. Rigorous data cleaning includes removing missing values, correcting erroneous codes, and standardizing dates and billing codes to ensure reliable pretraining data. The following sections detail how we construct patient medical histories and preprocess data to create the cancer screening benchmark.

**Data De-identification and Privacy.** All NHIRD data are de-identified following strict protocols mandated by the National Health Insurance Administration (NHIA). Personally identifiable information (PII), such as names and ID numbers, is removed and irreversibly encrypted using non-public anonymization methods (Lin et al., 2018; Health & Center). Our study uses only anonymized demographic variables, e.g., age and gender, for model training. Given the level of anonymization and the coarse granularity of these attributes, the risk of patient re-identification is negligible. Hence, our use of NHIRD data fully complies with privacy standards and poses no ethical or legal concerns regarding patient confidentiality.

**Correctness of NHIRD.** The universal healthcare system in Taiwan enables frequent patient visits and interactions and generates rich, detailed, and longitudinal patient records. In 2023, NHIRD recorded over 380 million medical visits for 23 million individuals, averaging 16.5 visits per person (tai, 2024). These statistics, including consistent surgery and prescription rates, align closely with our dataset statistics in Table 1, confirming the correctness of our dataset. Most importantly, all data in NHIRD receive strict validation by the National Health Insurance Administration (NHIA), which enforces robust quality control measures to eliminate duplication, correct inconsistencies, and ensure patient-level accuracy. Additionally, coding standards are uniformly implemented across all healthcare providers nationwide, guaranteeing consistency and correctness at scale. Given this systematic validation, national standardization, and population-wide coverage, NHIRD is an exceptionally accurate and dependable data source trusted and used in research published at top peer-reviewed medical journals (Wang et al., 2018a; Lee et al., 2019; Tsai et al., 2024; Tain et al., 2025).

**Cancer Diagnosis Validity.** Cancer diagnoses in the NHIRD are made by licensed physicians from all hospitals and medical facilities and undergo strict validation by the National Health Insurance Administration (NHIA) to ensure diagnostic accuracy, prevent misclassification, and eliminate billing errors or fraud (Lin et al., 2018; Hsieh et al., 2019). This rigorous quality control process makes the NHIRD a highly reliable source for cancer retrospective research, especially for creating cancer patient cohorts with diagnosis codes. With verified patient histories and clinically validated cancer diagnoses, the NHIRD has been widely adopted for cancer research and trusted in top peer-reviewed studies (Lin et al., 2015; Chien et al., 2016; Huang et al., 2023).

**Medical History Construction.** To create a sequential medical history for each patient, we first aggregate all visits and their associated order details by joining CD with OO, DD with DO, and GD with GO. Figure 6a describes the join process on CD with OO. After that, we aggregate visits for each individual 4 million patients by their patient ID and sorted in chronological order. Figure 6b shows a sample patient after the aggregation on all visits.

**Cancer Demographic Statistics.** Table 10 presents demographic statistics, including cancer patients' medical history lengths across age groups. The NHIRD provides extensive longitudinal data, with average history lengths ranging from 10.3 to 15 years across various cancers. This depth allows for analyzing disease progression and identifying patterns that support early cancer detection. Its large sample size ensures robust statistical power for subgroup analysis, making NHIRD an invalu-

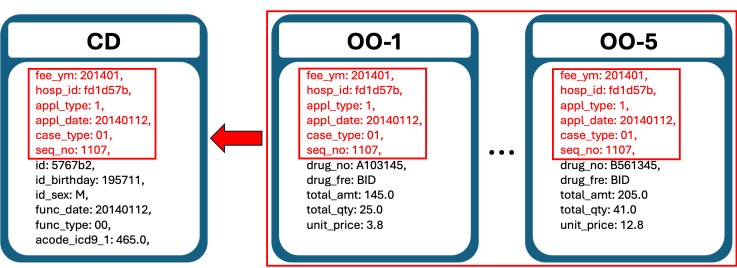

(a) Sample table joins on CD with OO, assuming a totally 5 order details. All values for each field are synthetic to maintain PIIs. We join CD with OO on **fee_ym**, **hosp_id**, **appl_type**, **appl_date**, **case_type**, and **seq_no**. The joining between DD with DO and GD with GO follows the same methods described here.

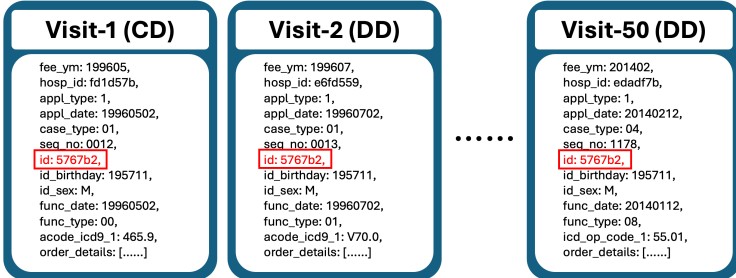

(b) Sample patient medical history, assuming a totally 50 visits, after aggregating all CD, DD, and GD by patient **id**. We differentiate visit types by their original table before aggregation.

Figure 6: Sample join and resulting example patient.

Table 8: NHIRD table overview. Table name with * denotes the table we use in our paper.

| Name | Description | Name | Description |
|---|---|---|---|
| BED | Registry for contracted beds | ID* | Registry for beneficiaries (Patient information) |
| DETA | Registry for contracted specialty services | DT | Monthly claim summary for inpatient claims |
| HOSB | Registry for contracted medical facilities | CT | Monthly claim summary for ambulatory care claims |
| HOSX | Supplementary registry for contracted medical facilities | DD* | Inpatient expenditures by admissions (Hospitalization visit) |
| DOC | Registry for board-certified specialists | DO* | Details of inpatient orders (Order detail of Hospitalization visit) |
| PER | Registry for medical personnel | CD* | Ambulatory care expenditures by visits (Outpatient clinic visits) |
| HV | Registry for catastrophic illness patients | OO* | Details of ambulatory care orders (Order detail of outpatient clinic visits) |
| HOX | Registry for medical services | GD* | Expenditures for prescriptions dispensed at contracted pharmacies (Pharmacy visit) |
| DRUG | Registry for drug prescriptions | GO* | Details of prescriptions dispensed at contracted pharmacies (Order detail of pharmacy visit) |

Table 9: Demographic, history length, and other statistics of NHIRD. Note that, the average and median history length are calculated in years. Notably, some patient gender records are not male or female, so we exclude those from the statistics.

| Group | Count | Avg. history | Median history | Avg. # visits | Median # visits | Avg. # codes | Median # codes |
|---|---|---|---|---|---|---|---|
| All | 3,989,369 | 15.2 | 17 | 271 | 214 | 5,886 | 4,329 |
| Male | 1,965,368 | 15.1 | 17 | 243 | 184 | 5,415 | 3,818 |
| Female | 1,962,523 | 16 | 17 | 306 | 248 | 6,504 | 4,934 |
| 0-18 | 456,322 | 12.4 | 13 | 274 | 250 | 5,803 | 5,208 |
| 18-35 | 1,047,623 | 15.4 | 17 | 188 | 164 | 3,736 | 3,137 |
| 35-50 | 981,514 | 15.5 | 17 | 214 | 172 | 4,448 | 3,384 |
| 50-70 | 1,017,522 | 16 | 17 | 312 | 253 | 6,829 | 5,246 |
| 70+ | 486,388 | 15 | 17 | 477 | 416 | 11,525 | 9,871 |

able resource for population-wide studies on cancer progression, early detection, and screening effectiveness.

**Subsequent Cancer Definition.** In cancer registry standards such as SEER and IARC, the term *subsequent primary cancer* refers strictly to new independent primary malignancies, explicitly excluding metastases or recurrences of prior cancers. In contrast, for the purposes of this study, we define *subsequent cancer* more broadly to include both new independent primary cancers and metastases to the target organ. This operational definition reflects the screening context, where both scenarios represent clinically relevant risks for early detection.

## B    DATA ACCESSIBILITY, IRB, AND REPRODUCIBILITY

NHIRD is a publicly accessible research resource governed under controlled access. Any research institution in Taiwan may apply for access by obtaining Institutional Review Board (IRB) approval and complying with the data use regulations set by the Ministry of Health and Welfare. International research institutions can collaborate with Taiwanese research institutions under a formal agreement. While the data are not openly accessible, they are available to all eligible institutions through a formal application process. We obtained NHIRD access through a formal collaboration with a Taiwanese medical school. As part of the IRB review process, we submitted the complete NHIRD data schema, a comprehensive research plan, the data access protocol, and a list of authorized users. We received formal approval and granted exemption from our IRB, as the study relied solely on fully de-identified secondary data and involved no human subjects. Such IRB determination and exemption are common in research using the MIMIC dataset. For reproducibility, we will publicly release our complete data preprocessing pipeline and modeling code. In addition, pre-trained model checkpoints are made available to researchers who obtain NHIRD access through the same framework, enabling fair replication and downstream research while upholding strict privacy and compliance standards.

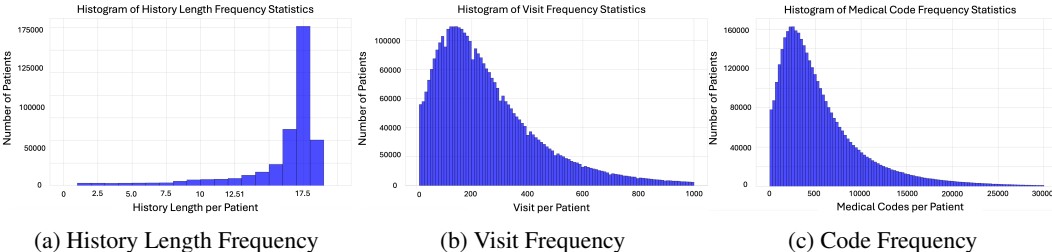

(a) History Length Frequency  (b) Visit Frequency  (c) Code Frequency

Figure 7: NHIRD frequency statistics on history length, number of visit, and number of codes per patient

Table 10: Demographics and medical history lengths of selected cancer patients in NHIRD. The difference in cancer case counts from Table 2 results from the removal of cases with insufficient history or erroneous records during benchmark creation.

| Demographic group | Counts | Avg. history length |
|---|---|---|
| **Pancreatic** | | |
| All | 12,602 | 12.3 |
| 0-18 | 28 | 11.8 |
| 18-35 | 267 | 15 |
| 35-50 | 1,453 | 13.3 |
| 50-70 | 4,806 | 12.8 |
| 70+ | 6,048 | 11.6 |
| **Liver** | | |
| All | 11,355 | 13.7 |
| 0-18 | 14 | 10.3 |
| 18-35 | 252 | 14.3 |
| 35-50 | 1,620 | 14.2 |
| 50-70 | 4,968 | 13.9 |
| 70+ | 4,501 | 13.1 |
| **Lung** | | |
| All | 8,867 | 14 |
| 0-18 | 15 | 12.8 |
| 18-35 | 120 | 14.5 |
| 35-50 | 678 | 14.3 |
| 50-70 | 3,109 | 14.4 |
| 70+ | 4,945 | 13.8 |

## C   EHRSHOT

**Dataset and Prediction Task Overview.** EHRSHOT is the latest published EHR dataset for evaluating the few-shot performance of foundation models on clinical prediction tasks. It consists of structured, de-identified, and longitudinal Electronic Health Records (EHRs) from 6,739 patients treated at Stanford Medicine, including diagnosis codes, procedures, medications, and laboratory test results. EHRSHOT is sourced from the EHR system in both Stanford Health Care and Lucile Packard Children's Hospital, where 2.57M patients used to pretrain CLMBR, a clinical foundation model used to evaluate the EHRSHOT benchmark in the original paper (Wornow et al., 2023). It contains 15 distinct clinical prediction tasks under the four main categories:

- Operational Outcomes

- Anticipating Lab Test Results

- Assignment of New Diagnoses

- Anticipating Chest X-ray Findings

Table 11: Comparison of coding formats between EHRSHOT and NHIRD, where NHI refers to Taiwan's National Health Insurance Administration. In NHIRD, only ICD-9-CM and ICD-9-Proc are international standard codes for diagnoses and procedures, while all other codes, such as those for drugs, orders, materials, and services, are defined by the NHI for domestic use only.

| Code type | EHRSHOT | NHIRD |
|---|---|---|
| Disgnosis | SNOMED, ICDO3 | ICD-9-CM |
| Prescriptions | RxNorm | NHI Drug Code |
| Medical Procedures | CPT, HCPCS | NHI Order Code |
| Surgical Procedures | SNOMED, ICD-10-Proc, ICD-9-Proc, | ICD-9-Proc, NHI Order Code |
| Diagnositc Procedures | CPT, HCPCS | NHI Order Code |
| Lab Orders | LONIC | NHI Order Code |
| Medical Equipments | SNOMED, CPT, HCPCS | NHI Material Code |
| Medical Supplies | HCPCS | NHI Material Code |
| Medical Services | CPT | NHI Service Code |

Table 12: Summary of coding mapping from EHRSHOT to NHIRD.

| Code in EHRSHOT | Mapping methods | Code in NHIRD |
|---|---|---|
| SNOMED | SNOMED to ICD9 mapping | ICD-9-CM |
| SNOMED - procedure | Semantic text matching | ICD-9-Proc, NHI Order Code |
| SNOMED - regime/therapy | Semantic text matching | ICD-9-Proc, NHI Order Code |
| SNOMED - physical object | Semantic text matching | NHI Order Code |
| ICDO3 | ICD mapping by CMS | ICD-9-CM |
| ICD10-Proc | ICD mapping by CMS | ICD-9-Proc |
| RxNorm | Semantic text matching | NHI Drug Code |
| CPT | Semantic text matching | NHI Order / Service Code |
| HCPCS | Semantic text matching | NHI Material Code |
| LONIC | Semantic text matching | NHI Order Code |

where the one-year pancreatic cancer prediction is under the category "Assignment of New Diagnoses". It is worth noting that, pancreatic cancer prediction is the only available cancer prediction task in EHRSHOT. Each designed to test the ability of foundation models for accurate predictions under limited labeled data at the time of patient visit. Refer Table 3 in the original EHRSHOT paper (Wornow et al., 2023) for complete task description and statistics.

**EHRSHOT verse NHIRD.** The source EHR system of EHRSHOT in Stanford Medicine and Lucile Packard Children's Hospital utilizes the Observational Medical Outcomes Partnership Common Data Model (OMOP-CDM) format. The OMOP-CDM is a standardized format for organizing and conforming EHR, enabling consistent analysis across diverse healthcare datasets. On the other hand, The NHIRD, as described in Appendix A, comprises an extensive set of structured tables capturing nearly all aspects of healthcare encounters. However, its format is tailored for internal use by the National Health Insurance Administration in Taiwan, rather than for international interoperability like OMOP-CDM. Applying CATCH-FM, built and pretrained on NHIRD, to EHRSHOT is nontrivial due to differences in coding formats between the two healthcare systems. Table 11 summarizes the key differences in medical coding schemes used by NHIRD and EHRSHOT.

**Mapping EHRSHOT to NHIRD.** As CATCH-FM is pretrained on NHIRD using ICD-9 and NHI codes, we map SNOMED to ICD-9 using official CMS and NLM mappings. While 43% of codes have one-to-one mappings (mostly SNOMED to ICD-9), the remaining 57% (e.g., RxNorm, CPT) are aligned to NHI codes via semantic matching on text descriptions (Liu et al., 2020; Sung et al., 2020a; Yuan et al., 2022). Due to differences in drug, procedure, and medical codes, the average cosine similarity between EHRSHOT and top-1 matched NHIRD codes is 84.3%, posing a significant generalization challenge where no gold-mapping standard exists.

ICD-based codes such as ICD-9-Proc (ICD-9 Procedure code), ICD-10-PCS (ICD-10 Procedure code), and ICD-O-3 (ICD oncology code) can be mapped to ICD-9 using publicly available mappings from the Centers for Medicare & Medicaid Services (CMS). Diagnostic concepts in SNOMED can

Table 13: Summary of coding mapping from EHRSHOT to NHIRD.

| Code in EHRSHOT | Mapping methods | Code in NHIRD |
|---|---|---|
| SNOMED | SNOMED to ICD9 mapping | ICD-9-CM |
| SNOMED - procedure | Semantic text matching | ICD-9-Proc, NHI Order Code |
| SNOMED - regime/therapy | Semantic text matching | ICD-9-Proc, NHI Order Code |
| SNOMED - physical object | Semantic text matching | NHI Order Code |
| ICDO3 | ICD mapping by CMS | ICD-9-CM |
| ICD10-Proc | ICD mapping by CMS | ICD-9-Proc |
| RxNorm | Semantic text matching | NHI Drug Code |
| CPT | Semantic text matching | NHI Order / Service Code |
| HCPCS | Semantic text matching | NHI Material Code |
| LONIC | Semantic text matching | NHI Order Code |

Table 14: Code mapping statistics from EHRSHOT to NHIRD. "Exact" denotes mappings via official mapping, while "Threshold" refers to semantic text matching at the specified cosine similarity (0.0–1.0). The 0.98 soft-matching cutoff is applied only to codes without official mappings. The 0.98 threshold includes both exact and those mapped by the 0.98 soft-matching cutoff.

| Code type (Total) | Exact | 0.98 threshold |
|---|---|---|
| SNOMED (11598) | 8998 (77.6%) | 9072 (78.2%) |
| ICD10PCS (3669) | 3618 (98.6%) | 3618 (98.6%) |
| ICD03 (96) | 76 (79.2%) | 83 (86.5%) |
| RxNorm (5433) | 0 (0.0%) | 0 (0.0%) |
| CPT (4675) | 0 (0.0%) | 3 (0.06%) |
| LONIC (3945) | 2 (0.05%) | 3 (0.08%) |
| HCPCS (64) | 0 (0.0%) | 0 (0.0%) |
| All (29480) | 12694 (43.1%) | 12779 (43.4%) |

also be mapped to ICD-9 via the "ICD-9-CM Diagnostic Codes to SNOMED CT Map" provided by the U.S. National Library of Medicine (NLM). However, other coding systems, e.g., SNOMED procedures, CPT (coding for medical services, procedures, and other practices), and RxNorm (codes for drug prescriptions), lack direct mappings to the NHIRD coding scheme. Table 13 summarizes the mapping strategies and source-target relationships between each coding system. To address this, we adopt a text-based semantic matching approach by embedding[1] the textual descriptions of codes using Sentence Transformers (Thakur et al., 2021) and formulating the mapping as a dense retrieval problem, a method widely used to align medical concepts across ontologies (Sung et al., 2020a; Liu et al., 2020; Yuan et al., 2022). Codes from each system are matched to the most semantically similar code in NHIRD based on text similarity, using Faiss as the retrieval backend (Douze et al., 2024).

Since no gold mapping standard exists for semantic alignment, it is difficult to determine whether moderate similarity scores (e.g., 0.7–0.85) reflect true mappings or noise. To approximate exact mappings and ensure high precision, we therefore adopt a strict cutoff of 0.98 as the threshold for soft matching. Table 14 summarizes the resulting coverage under different matching strategies. Even with semantic matching, the final coverage reaches only 43%, implying that over half of the medical information is lost. This substantial loss reflects the inherent challenge of aligning heterogeneous and non-standardized EHR format, especially when deploying healthcare foundation models across different healthcare systems and populations, and underscores the difficulties of achieving robust model generalization.

## D  IMPLEMENTATION DETAILS

**Tokenization and inputs.** We map all medical codes to unique indices ranging from 0 to the total number of unique medical codes, with a token vocabulary size of 185,138, including demographic, time, and special tokens. Since all tokens are treated as atomic units, no additional tokenization

---

[1]We adopt Salesforce/SFR-Embedding-Mistral as encoder from Huggingface.

Table 15: Model hyperparameters under different parameter sizes with million (m) and billion (b).

| Parameters | Num of layers | Dimension | Num of heads | Block size |
|---|---|---|---|---|
| 70m | 6 | 512 | 8 | 2048 |
| 120m | 6 | 768 | 8 | 2048 |
| 160m | 12 | 768 | 12 | 2048 |
| 260m | 12 | 1024 | 16 | 2048 |
| 350m | 20 | 1024 | 16 | 2048 |
| 410m | 24 | 1024 | 16 | 2048 |
| 560m | 22 | 1280 | 10 | 2048 |
| 720m | 20 | 1536 | 12 | 2048 |
| 1b | 16 | 2048 | 8 | 2048 |
| 1.2b | 20 | 2048 | 16 | 2048 |
| 1.4b | 24 | 2048 | 16 | 2048 |
| 2.1b | 24 | 2560 | 16 | 2048 |
| 2.8b | 32 | 2560 | 32 | 2048 |

Table 16: Hyperparameters Configurations for CATCH-FM

| Hyperparameter | Pretraining | Supervised Fine-tuning |
|---|---|---|
| Learning Rate | \multicolumn{2}{c}{6e-6 for model size $\geq$ 1B; else 1e-5} | |
| Optimizer | AdamW | |
| Adam $\epsilon$ | 1e-8 | |
| Adam Betas ($\beta_1$, $\beta_2$) | (0.9, 0.999) | |
| Weight decay | 0.01 | |
| Gradient Norm | 0.1 | |
| Scheduler | Warmup-Stable-Decay | |
| Warmup Ratio | 0.1 | |
| Stable Ratio | 0.8 | |
| Decay Ratio | 0.1 | |
| Batch Size | 64 | 128 |
| Epochs | - | 5 |

is required. The input sequence length is limited to 2,048 tokens. For patient records with EHR sequences exceeding this limit, we avoid truncation during pretraining and instead split the sequences into non-overlapping chunks, processing them across multiple training steps. However, during fine-tuning, inputs longer than 2,048 tokens are truncated, as they must be processed as single sequences.

**Backbone architecture.** We use the Pythia architecture as the backbone of CATCH-FM. Pythia is a family of decoder-only autoregressive language models, ranging from 70m to 12b parameters, designed for scalable and consistent research. Its feedforward architecture features rotary embeddings for positional encoding, untied embedding layers, and parallelized flash attention for efficient training. Table 15 outlines the model architecture, while Table 16 details the hyperparameters for pretraining and supervised fine-tuning on cancer screening tasks. All models are trained on 8 A100-SXM4-40GB GPUs.

**Baselines.** For the tree-based baselines, XGBoost and LightGBM, we utilize their official Python packages, xgboost [2] and lightgbm [3], respectively. For deep learning baselines, we employ the PyHealth framework (Yang et al., 2023a) and perform supervised learning tasks directly on the cancer screening benchmarks. For language model baselines, we convert medical codes into text using language model vocabularies and concatenate them into sequences representing patient histories for fine-tuning. In NHIRD, ICD9 codes are mapped to standardized textual descriptions using official code mapping tables. Each medical code, representing a diagnosis, surgery, treatment, or medication,

---

[2]https://xgboost.readthedocs.io/en/stable/index.html
[3]https://lightgbm.readthedocs.io/en/stable/

Table 17: Estimated FLOPs requirement for CATCH-FM. The required training steps and total tokens (inside the parentheses) for different model sizes are reported here.

| FLOPS | 1e+18 | 4e+18 | 8e+18 | 2e+19 | 4e+19 | 8e+19 | 1e+20 | 2e+20 |
|---|---|---|---|---|---|---|---|---|
| CATCH-FM-70m | 10071 (1.32e+09) | 40287 (5.28e+09) | 80575 (1.06e+10) | 201439 (2.64e+10) | 402878 (5.28e+10) | 805757 (1.06e+11) | 1007197 (1.32e+11) | 2014394 (2.64e+11) |
| CATCH-FM-120m | 6247 (8.19e+08) | 24990 (3.28e+09) | 49981 (6.55e+09) | 124952 (1.64e+10) | 249905 (3.28e+10) | 499810 (6.55e+10) | 624763 (8.19e+10) | 1249526 (1.64e+11) |
| CATCH-FM-160m | 4800 (6.29e+08) | 19202 (2.52e+09) | 38405 (5.03e+09) | 96014 (1.26e+10) | 192029 (2.52e+10) | 384058 (5.03e+10) | 480072 (6.29e+10) | 960145 (1.26e+11) |
| CATCH-FM-260m | 3252 (4.26e+08) | 13011 (1.71e+09) | 26022 (3.41e+09) | 65057 (8.53e+09) | 130114 (1.71e+10) | 260228 (3.41e+10) | 325285 (4.26e+10) | 650570 (8.53e+10) |
| CATCH-FM-350m | 2421 (3.17e+08) | 9685 (1.27e+09) | 19371 (2.54e+09) | 48429 (6.35e+09) | 96858 (1.27e+10) | 193716 (2.54e+10) | 242145 (3.17e+10) | 484290 (6.35e+10) |
| CATCH-FM-410m | 2147 (2.81e+08) | 8588 (1.13e+09) | 17176 (2.25e+09) | 42941 (5.63e+09) | 85882 (1.13e+10) | 171765 (2.25e+10) | 214706 (2.81e+10) | 429412 (5.63e+10) |
| CATCH-FM-720m | 1302 (1.71e+08) | 5209 (6.83e+08) | 10418 (1.37e+09) | 26045 (3.41e+09) | 52090 (6.83e+09) | 104180 (1.37e+10) | 130225 (1.71e+10) | 260451 (3.41e+10) |
| CATCH-FM-1b | 964 (1.26e+08) | 3857 (5.06e+08) | 7714 (1.01e+09) | 19285 (2.53e+09) | 38570 (5.06e+09) | 77141 (1.01e+10) | 96426 (1.26e+10) | 192853 (2.53e+10) |
| CATCH-FM-1.2b | 818 (1.07e+08) | 3273 (4.29e+08) | 6547 (8.58e+08) | 16369 (2.15e+09) | 32739 (4.29e+09) | 65478 (8.58e+09) | 81848 (1.07e+10) | 163696 (2.15e+10) |
| CATCH-FM-1.4b | 710 (9.32e+07) | 2843 (3.73e+08) | 5687 (7.46e+08) | 14219 (1.86e+09) | 28439 (3.73e+09) | 56879 (7.46e+09) | 71098 (9.32e+09) | 142197 (1.86e+10) |
| CATCH-FM-2.1b | 486 (6.38e+07) | 1946 (2.55e+08) | 3892 (5.10e+08) | 9732 (1.28e+09) | 19464 (2.55e+09) | 38929 (5.10e+09) | 48662 (6.38e+09) | 97324 (1.28e+10) |
| CATCH-FM-2.8b | 382 (5.01e+07) | 1529 (2.00e+08) | 3058 (4.01e+08) | 7646 (1.00e+09) | 15292 (2.00e+09) | 30584 (4.01e+09) | 38230 (5.01e+09) | 76460 (1.00e+10) |

Table 18: Operational Decision Threshold Analysis of CATCH-FM-2.4b on *first* target cancer cohorts

| Threshold | False Positive Rate | True Positive Rate | Specificity | Precision | Relative Risk |
|---|---|---|---|---|---|
| **Cancer**: Pancreatic, **Positive/Negative (Incidence Ratio)**: 452/28058 (1.59%) | | | | | |
| 0.996 | 0.000 | 0.188 | 1.000 | 1.000 | 63.075 |
| 0.980 | 0.000 | 0.330 | 0.9999 | 0.974 | 61.426 |
| 0.932 | 0.001 | 0.407 | 0.9994 | 0.911 | 57.455 |
| 0.815 | 0.001 | 0.458 | 0.9986 | 0.841 | 53.075 |
| 0.686 | 0.002 | 0.491 | 0.9975 | 0.760 | 47.954 |
| 0.495 | 0.004 | 0.535 | 0.9960 | 0.684 | 43.119 |
| 0.411 | 0.005 | 0.566 | 0.9946 | 0.627 | 39.577 |
| 0.264 | 0.008 | 0.586 | 0.9918 | 0.536 | 33.836 |
| 0.204 | 0.010 | 0.606 | 0.9900 | 0.495 | 31.196 |
| **Cancer**: Liver, **Positive/Negative (Incidence Ratio)**: 509/26148 (1.95%) | | | | | |
| 0.984 | 0.000 | 0.134 | 0.9999 | 0.958 | 50.158 |
| 0.957 | 0.001 | 0.242 | 0.9994 | 0.891 | 46.679 |
| 0.905 | 0.001 | 0.330 | 0.9985 | 0.816 | 42.711 |
| 0.841 | 0.003 | 0.381 | 0.9972 | 0.727 | 38.053 |
| 0.768 | 0.004 | 0.422 | 0.9958 | 0.660 | 34.539 |
| 0.702 | 0.006 | 0.446 | 0.9945 | 0.612 | 32.044 |
| 0.635 | 0.007 | 0.477 | 0.9932 | 0.576 | 30.157 |
| 0.588 | 0.008 | 0.501 | 0.9919 | 0.545 | 28.536 |
| 0.515 | 0.010 | 0.536 | 0.9901 | 0.512 | 26.824 |
| **Cancer**: Lung, **Positive/Negative (Incidence Ratio)**: 868/30153 (1.45%) | | | | | |
| 0.997 | 0.000 | 0.130 | 0.9998 | 0.897 | 62.455 |
| 0.992 | 0.001 | 0.243 | 0.9995 | 0.866 | 60.332 |
| 0.977 | 0.001 | 0.336 | 0.9992 | 0.854 | 59.456 |
| 0.950 | 0.002 | 0.366 | 0.9982 | 0.749 | 52.157 |
| 0.914 | 0.003 | 0.414 | 0.9974 | 0.702 | 48.911 |
| 0.825 | 0.004 | 0.454 | 0.9962 | 0.633 | 44.087 |
| 0.716 | 0.005 | 0.491 | 0.9950 | 0.589 | 41.002 |
| 0.593 | 0.007 | 0.509 | 0.9931 | 0.519 | 36.167 |
| 0.385 | 0.010 | 0.531 | 0.9901 | 0.439 | 30.576 |

is transformed accordingly. Patient histories are then constructed by concatenating the medical text, similar to processing text documents in language models. The sequence length limit is 1024 for BioGPT and 2048 for Qwen.

**Training Configurations.** Table 17 shows the required training steps and total tokens for CATCH-FM at various FLOP targets. We adopt this configuration to investigate the scaling laws for CATCH-FM, aiming to determine the optimal model and data scales under a fixed FLOP budget. Their corresponding FLOPs, number of parameters, and GPU training hours are listed in Table 3. We calculate the FLOPs with Pytorch built-in flops counter.

# E   ANALYSIS OF THRESHOLD TUNING

Clinically, models often adjust decision thresholds to prioritize sensitivity or specificity (Collins & Moons, 2012; Steyerberg & Steyerberg, 2019), such as reducing false positives in prescreening or increasing sensitivity in high-risk groups. Figure 8 shows that CATCH-FM supports tunable sensitivity and specificity, which makes it well-suited for diverse clinical settings. It show that CATCH-FM maintains its advantage across all thresholds, confirming its ability to identify high-risk patients for further screening (high sensitivity) and avoid unnecessary patient distress (high specificity). We also report AUROC curves across thresholds to compare our model with baseline methods (XGBoost and Qwen), as shown in Figure 9.

Table 19: Model Performance Comparsion (Specificity = 0.99) on *first* target cancer cohort

| Model (Input) | Threshold | False Positive Rate | True Positive (Sensitivity) | Specificity | Precision | Relative Risk |
|---|---|---|---|---|---|---|
| **Cancer**: Pancreatic, **Positive/Negative (Incidence Ratio)**: 452/28058 (1.59%) | | | | | | |
| CATCH-FM-2.4b (Code Sequence) | 0.204 | 0.01 | 0.606 | 0.99 | 0.495 | 31.196 |
| XGBoost (Bag-of-words) | 0.180 | 0.01 | 0.310 | 0.99 | 0.335 | 21.126 |
| Qwen (Language Sequence) | 0.408 | 0.01 | 0.254 | 0.99 | 0.324 | 17.644 |
| **Cancer**: Liver, **Positive/Negative (Incidence Ratio)**: 509/26148 (1.95%) | | | | | | |
| CATCH-FM-2.4b (Code Sequence) | 0.515 | 0.01 | 0.536 | 0.99 | 0.512 | 26.824 |
| XGBoost (Bag-of-words) | 0.200 | 0.01 | 0.363 | 0.99 | 0.415 | 21.723 |
| Qwen (Language Sequence) | 0.437 | 0.01 | 0.324 | 0.99 | 0.421 | 19.3 |
| **Cancer**: Lung, **Positive/Negative (Incidence Ratio)**: 868/30153 (1.45%) | | | | | | |
| CATCH-FM-2.4b (Code Sequence) | 0.385 | 0.01 | 0.531 | 0.99 | 0.439 | 30.576 |
| XGBoost (Bag-of-words) | 0.172 | 0.01 | 0.323 | 0.99 | 0.323 | 22.5 |
| Qwen (Language Sequence) | 0.573 | 0.01 | 0.188 | 0.99 | 0.238 | 15.239 |

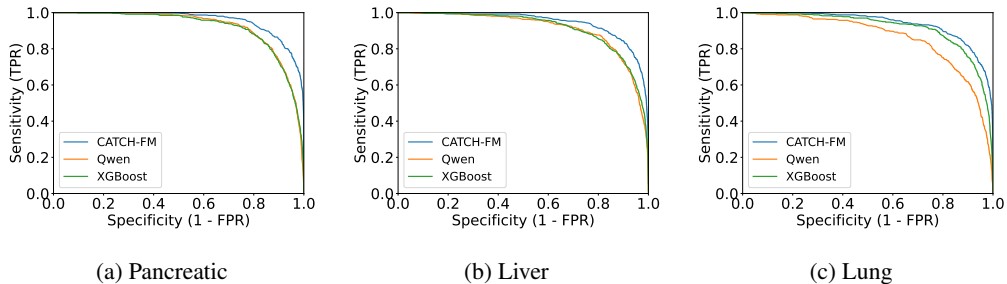

(a) Pancreatic          (b) Liver          (c) Lung

Figure 8: Sensitivity and specificity as functions of the screening decision threshold for CATCH-FM-2.4b, Qwen2.5-500M, and XGBoost on the *first* target cancer cohorts.

We evaluate our model and carefully select decision thresholds from the AUROC curve to ensure a specificity of at least 0.99. We also report Relative Risk (RR), to quantify the odds of cancer among those classified as positive compared to a random pick based just on the population disease incidence following suggested work in Placido et al. (2023a). We provide a complete threshold selection results in Table 18.

Under a fixed specificity of 0.99 and FPR of 0.01, our model demonstrates strong performance across all three cancer prediction tasks. For pancreatic cancer, a threshold of 0.20 yields a TPR of 0.61, and a relative risk of 31.2. For liver cancer, a threshold of 0.52 yields a TPR of 0.54 and a relative risk of 26.8. For lung cancer, a threshold of 0.39 yields a TPR of 0.53 and a relative risk of 30.6. These results highlight the utility of our method across diverse cancer types. Under the fixed specificity threshold, we ensure a fair comparison with the baselines, as detailed in Table 19. Across all metrics, our model consistently outperforms the baselines.

To simulate clinical implementation, following the approach proposed in Placido et al. (2023a), we adopt an operational decision point that simulates cost constraints, where only the top 0.1% of patients (by predicted risk) are eligible to be advanced to a surveillance program. At this threshold, our model achieves relative risk scores of 63.1, 52.0, and 61.9 for pancreatic, liver, and lung cancers prediction, respectively. This also further indicates the clinical utility of our model.

# F    BENEFITS OF COMPUTE-OPTIMAL PRETRAINING

We show the detailed performance comparison between compute-(non-)optimal pretraining on downstream cancer pancreatic screening, as supplementary information for Figure 2, in the Table 20. We can observe that a compute-optimal model with adequate tokens outperforms larger models with insufficient tokens and smaller models with excessive tokens.

# G    ANALYSIS OF COHORT CONTROL

We evaluate CATCH-FM-160m's performance with training and testing on controlled (matched) and random (out-of-distribution) control groups to assess screening under various distribution shifts.

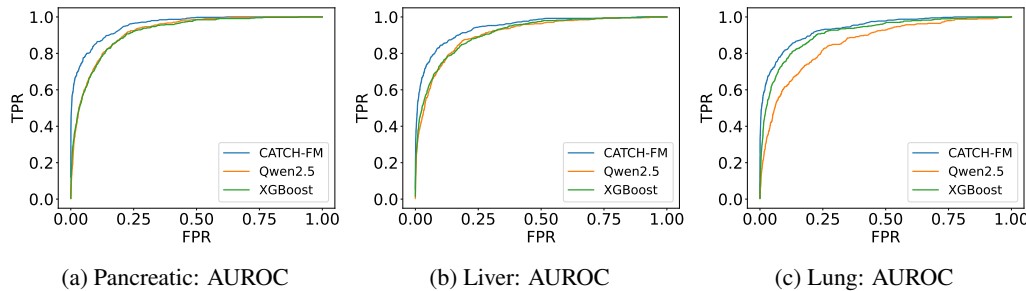

(a) Pancreatic: AUROC      (b) Liver: AUROC      (c) Lung: AUROC

Figure 9: AUROC comparison across cancer cohorts with CATCH-FM-2.4b, Qwen2.5-500M, and XGBoost evaluated on the *first* target cancer cohorts.

Table 20: Performance of compute-(non-)optimal pretraining on *all* target cancer cohorts.

| Scale | Tokens | Loss | F1 | AUROC | AUPRC | Specificity | Sensitivity |
|---|---|---|---|---|---|---|---|
| **FLOPs = 4e18, Optimal Model Scale = 260m** | | | | | | | |
| 260m | 1.7B | 2.887 | 86.6 | 97.1 | 81.2 | 99.5 | 70.7 |
| 70m | 5.2B | 2.974 | 86.4 | 97.0 | 81.1 | 99.2 | 70.7 |
| 2b | 0.2B | 3.411 | 82.7 | 96.1 | 73.3 | 99.1 | 60.9 |
| **FLOPs = 2e19, Optimal Model Scale = 410m** | | | | | | | |
| 410m | 5.6B | 2.677 | 87.3 | 96.4 | 81.2 | 99.3 | 70.8 |
| 70m | 26B | 2.862 | 86.1 | 96.2 | 79.7 | 99.1 | 70.7 |
| 2b | 1.2B | 2.885 | 82.4 | 96.2 | 74.9 | 98.5 | 67.3 |

Table 21: Evaluation of pretrained compute-optimal model CATCH-FM-160m on *all* cancer cohorts with different target controls on negative cases. The label distribution is fixed for each cancer data.

| Negative Selection | | F1 (Macro) | AUROC | AUPRC | Specificity | Sensitivity |
|---|---|---|---|---|---|---|
| Training | Testing | | | | | |
| **Cancer: Pancreatic** | | | | | | |
| Controlled | Controlled | 86.6 | 97.1 | 81.2 | 99.2 | 70.7 |
| Random | Random | 86.3 | 95.9 | 78.5 | 99.4 | 67.2 |
| Controlled | Random | 85.5 | 96.9 | 79.7 | 99.0 | 70.7 |
| Random | Controlled | 87.8 | 95.6 | 80.1 | 99.6 | 67.2 |
| **Cancer: Liver** | | | | | | |
| Controlled | Controlled | 84.5 | 96.0 | 76.6 | 99.3 | 63.4 |
| Random | Random | 85.0 | 95.0 | 75.8 | 99.3 | 64.5 |
| Controlled | Random | 79.4 | 94.6 | 69.0 | 98.2 | 63.4 |
| Random | Controlled | 85.2 | 94.6 | 75.5 | 99.4 | 64.5 |
| **Cancer: Lung** | | | | | | |
| Controlled | Controlled | 82.1 | 95.8 | 71.5 | 99.6 | 54.7 |
| Random | Random | 83.5 | 96.1 | 72.5 | 99.4 | 62.4 |
| Controlled | Random | 83.4 | 95.9 | 72.8 | 99.8 | 54.7 |
| Random | Controlled | 84.0 | 95.6 | 72.7 | 99.4 | 62.4 |

As shown in Table 21, CATCH-FM maintains strong performance on the random control group, demonstrating its robustness in handling out-of-distribution patients and ensuring consistent results across diverse populations.

Table 22: Performance of CATCH-FM-160m on all target cancer cohorts under varying exclusion windows.

| Time Window Exclusion | AUPRC | Specificity | Sensitivity |
|---|---|---|---|
| **Cancer**: Pancreatic | | | |
| 12-month | 81.2 | 99.2 | 70.7 |
| 6-month | 81.4 | 99.2 | 71.2 |
| **Cancer**: Liver | | | |
| 12-month | 76.6 | 99.3 | 63.4 |
| 6-month | 76.9 | 99.1 | 67.7 |
| **Cancer**: Lung | | | |
| 12-month | 71.5 | 99.6 | 54.7 |
| 6-month | 71.6 | 99.5 | 59.7 |

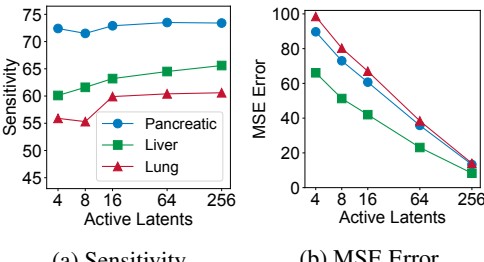

(a) Sensitivity     (b) MSE Error

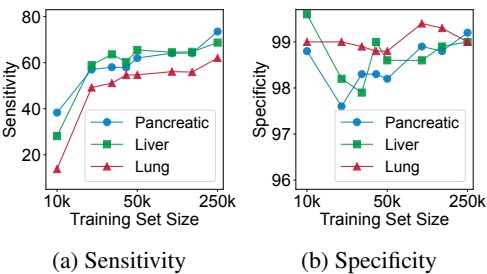

(a) Sensitivity     (b) Specificity

Figure 10: Evaluation results for the cancer screening task of the sparse autoencoder (SAE)

Figure 11: CATCH-FM-1b finetuned with different label sizes on *first* target cancer cohorts.

## H    TIME WINDOW EXCLUSION

This experiment studies CATCH-FM's effectiveness at different exclusion window in between input medical history and cancer diagnosis. Table 22 shows the results of CATCH-FM when finetuned and evaluated with different exclusion windows. A shortened 6-month exclusion window presents an easier task, as more short term risk factors may be observable, and CATCH-FM performs better in that setup. This confirms that CATCH-FM can be conveniently adapted to different prediction settings based on healthcare professional's preference.

## I    DETAILS OF INTERPRETABILITY EXPERIMENTS

We process positive patients' event token sequences using the fine-tuned CATCH-FM-1b model on each cancer dataset to obtain hidden states $h$ for every token. Following (Kang et al., 2024), we then train a TopK sparse autoencoder (SAE) on the hidden states of the [EOS] token, $h_{\texttt{[EOS]}}$, which serves as an aggregated representation of patient trajectories. The SAE is implemented as follows:

$$z = \mathrm{TopK}(W_{\mathrm{enc}}(h_{\texttt{[EOS]}} - b_{\mathrm{dec}}) + b_{\mathrm{enc}}), \tag{6}$$

$$\hat{h}_{\texttt{[EOS]}} = W_{\mathrm{dec}}z + b_{\mathrm{dec}} \tag{7}$$

where the embedding vector $h_{\texttt{[EOS]}}$ is passed through an encoder parameterized by $W_{\mathrm{enc}}$ and $b_{\mathrm{enc}}$. The TopK activation function regulates the number of active latent features. The encoded representation is then reconstructed via a decoder parameterized by $W_{\mathrm{dec}}$ and $b_{\mathrm{dec}}$. The SAE is trained using the mean squared error (MSE) loss for reconstruction.

Figure 10 presents the reconstruction evaluation across different numbers of active latent features on the cancer screening benchmark. We observe that as the number of active latent features increases, both cancer screening performance and reconstruction quality improve. Interestingly, with just 16 active latent features, the SAE's reconstructed embeddings already capture enough information

to match the original performance. This suggests that the latent cancer signal extracted from the fine-tuned CATCH-FM is inherently low-dimensional.

## J   ANALYSIS OF SUPERVISED DATA SCALE

Many healthcare systems may not have as many patients as in NHIRD, e.g., in scattered healthcare systems. This experiment evaluates CATCH-FM with different amounts of available supervised finetuning labels. Figure 11 plots CATCH-FM's performance finetuned with different amount of labels. CATCH-FM maintains its 99% specificity with as few as 10k training labels, with only 300 positives. Its sensitivity increases with more finetuning amount and crossed 50% with only 20k total patient data across two decades, which is fewer than a typical hospital.

## K   USAGE OF LARGE LANGUAGE MODEL

We used large language models to assist in polishing the writing of this paper. Specifically, LLMs were employed to correct grammar, paraphrase sentences, and improve readability and flow. The scientific ideas, experiments, and analyses were fully conducted by the authors, with LLM use limited to enhancing clarity and smoothness of expression.

