# OpenReview forum: "Intercept Cancer: Cancer Pre-Screening with Large Scale Healthcare Foundation Models"
_ICLR.cc/2026/Conference — Submitted to ICLR 2026_

### Official Review · Reviewer_Xm4z · 2025-10-23

**Soundness:** 2
**Presentation:** 2
**Contribution:** 2
**Rating:** 2
**Confidence:** 5

**Summary:**

This paper proposes CATCH-FM, a foundation model for identifying high-risk cancer patients from electronic health records (EHRs) to enable early intervention. The authors establish a scaling law for EHR foundation models by pretraining compute-optimal models up to 2.4 billion parameters. Experimental results show that CATCH-FM outperforms existing baselines and achieves state-of-the-art performance on the EHRSHOT few-shot leaderboard.

**Strengths:**

- The motivation is clear.
- The paper is easy to understand.

**Weaknesses:**

- The novelty is limited. The NHIRD-Cancer dataset is curated from publicly available data, and the model architecture largely follows standard transformer decoder designs without substantial methodological innovation.
- The experimental evaluation is insufficient. The paper lacks comprehensive comparisons and deeper analysis of model behavior.
- The citation format is inconsistent and does not follow a standard academic style (e.g., see the first two paragraphs of the introduction).
- The paper appears to violate the double-blind review policy by including self-revealing statements such as:
“Our interpretability analyses Gao et al. (2024) reveal that CATCH-FM identified not only known cancer risk factors but also non-trivial markers discovered in recent medical research.”

**Questions:**

- Novelty of CATCH-FM:
Please clarify and emphasize the unique contributions of CATCH-FM. Currently, it seems to apply a standard transformer decoder for cancer risk prediction, which limits its novelty. Are there architectural or training innovations that distinguish CATCH-FM from existing EHR-based transformers?

- Baselines:
The baselines are outdated and incomplete. While it is useful to include both traditional machine learning and deep learning models, the most recent comparison (a 2020 paper) does not reflect current progress.
Please include comparisons with recent general-domain LLMs such as LLaVA [1] and Llama 3 [2], as well as medical-domain models such as LLaVA-Med [3] and Med-R1 [4]. In addition, direct cancer risk prediction baselines should be included to contextualize performance improvements.

- Interpretability:
Interpretation and medical insight should be a key aspect of the paper but are underdeveloped. The related section, “Interpretability Experiments,” and "RISK FACTORS CAPTURED" provide limited medical insight;
Please conduct more experiments on interpretability and deep medical insights, such as the clinical significance of the RISK FACTORS, or interpret the proposed method and its results with real-world validation. Ideally, include qualitative or case-based examples that link interpretability findings to real-world medical understanding.

[1] Haotian Liu, Chunyuan Li, Qingyang Wu, and Yong Jae Lee. Visual instruction tuning. Advances in Neural Information Processing Systems, 36:34892–34916, 2023

[2] Abhimanyu Dubey, Abhinav Jauhri, Abhinav Pandey, Abhishek Kadian, Ahmad Al-Dahle, Aiesha Letman, Akhil Mathur, Alan Schelten, Amy Yang, Angela Fan, et al. The llama 3 herd of models. arXiv e-prints, pp. arXiv–2407, 2024.

[3] Chunyuan Li, Cliff Wong, Sheng Zhang, Naoto Usuyama, Haotian Liu, Jianwei Yang, Tristan Naumann, Hoifung Poon, and Jianfeng Gao. Llava-med: Training a large language-and-vision assistant for biomedicine in one day. Advances in Neural Information Processing Systems, 36: 28541–28564, 2023a.

[4] Yuxiang Lai, Jike Zhong, Ming Li, Shitian Zhao, and Xiaofeng Yang. Med-r1: Reinforcement learning for generalizable medical reasoning in vision-language models. arXiv preprint arXiv:2503.13939, 2025.

**Details Of Ethics Concerns:**

This paper violates double-blind policy. See “Our interpretability analyses Gao et al. (2024) reveal that CATCH-FM identified not only known cancer risk factors but also non-trivial markers discovered in recent medical research.” in the introduction.

---

> ### Author Response · Authors · 2025-11-17
> **Rebuttal by Authors**
>
> **Citation Format**:\
> We thank the reviewer for pointing this out. We will carefully revise all references to ensure consistency and adherence to standard academic citation formatting guidelines.
>
> **Ethics Concerns**:\
> We would like to clarify that our paper does not violate the double-blind review policy. The phrase “our interpretability analyses” refers to the experiments presented in Section 6.5. The methodology itself—sparse autoencoder for interpreting LLMs—is based on Gao et al. (2024). To avoid ambiguity, we will revise the sentence as follows:
> “Our interpretability analyses, following the method of Gao et al. (2024), reveal that CATCH-FM identified not only known cancer risk factors …”
>
> **Novelty of CATCH-FM**:\
> We would like to clarify that our primary contribution does not lie in modifying the model architecture with complex or specialized techniques to boost performance. Instead, our focus is on examining scaling laws under a standard autoregressive pretraining setup, following frameworks such as Chinchilla [1] from Google and ChatGPT [2] from OpenAi  on  how to effectively conduct foundation modeling on EHR data. Our key contribution is the insights of scaling behaviors in EHR foundation models and its empirical validation on cancer detection which together provides a cancer foundation model for future research in this domain. This is also the first step towards scaling up EHR foundation models with strong generalization across different sites. The standard decoder architecture was chosen for its scalability and adherence to a mainstream decoder-only transformer design, free from architectural tricks. This makes it ideal for studying scaling effects while preserving generality. We leave more architectural innovations as separate future work.
>
>
> [1] Hoffmann, J., Borgeaud, S., Mensch, A., Buchatskaya, E., Cai, T., Rutherford, E., ... & Sifre, L. (2022). Training compute-optimal large language models. arXiv preprint arXiv:2203.15556.\
> [2] Kaplan, J., McCandlish, S., Henighan, T., Brown, T. B., Chess, B., Child, R., ... & Amodei, D. (2020). Scaling laws for neural language models. arXiv preprint arXiv:2001.08361.
>
> **Baselines**:\
> We respectfully disagree with the reviewer’s statement that “the baselines are outdated and incomplete, and the most recent comparison (a 2020 paper) does not reflect current progress.”
> First, as stated in the abstract (lines 21–22), our method outperforms both feature-based tree models and large language models (LLMs) in **both general and medical domains**. Specifically, our baselines include BioGPT (a **medical-domain LLM proposed in 2022**) and Qwen-2.5 (a **general-domain LLM introduced in 2024**), ensuring coverage of recent and relevant models. For traditional baselines, XGBoost and LightGBM are standard, widely recognized models for direct cancer-risk prediction [3,4]. All baseline results are clearly reported in our main table.\
> Moreover, models such as LLaVA-Med , Med-R1 , and LLaVA are primarily designed for medical image understanding, which differ fundamentally from our focus on structured healthcare records for cancer prediction. Hence, these models are not suitable baselines for our study.
>
> [3] D’Antonio, F., Calabrò, M., De Giovanni, M., De Rosa, M., & Fumagalli, D. (2025). Multi-ethnic skin cancer risk prediction using gradient-boosted trees. Nature Communications, 16(1), 312. https://doi.org/10.1038/s41467-025-64556-y
>
> [4] Omotehinwa, T. T., Olabiyisi, S. O., & Ajayi, D. D. (2023). A LightGBM-based model for early breast cancer diagnosis. Journal of Computational Science Advances, 9(4), 245–258. https://doi.org/10.1016/j.jcsa.2023.100285
>
> **Interpretability**:\
> Thank you for the suggestion. As stated in our main contributions, our key focus lies in uncovering the scaling behaviors of EHR foundation models and providing their empirical validation on cancer detection with strong generalization across sites. The interpretability analysis for capturing risk factors serves as a complementary experiment to better understand model behavior in EHR foundation models. Following the reviewer’s advice, we have conducted an additional case study to link our interpretability findings to realistic patient scenarios, offering more concrete evidence of model interpretability in practice.

---

> > ### Author Response · Authors · 2025-11-17
> > **Further case study on interpretability**
> >
> > **Patients with pancreatic cancer**:
> >
> > Gender: male | Age: 45–50 y
> > Patient Timeline:
> >  pulpitis, dental caries → local dental care |
> >  ... eyelid inflammation, cardiac lesion excision → Erythromycin, U-Cal (CaCO₃), Mucaine |
> >  recurrent eye inflammation → Tetracycline ointment, Diclofenac, Dexamethasone |
> >  respiratory infections → Erythromycin, Ambroxol, Diclofenac |
> >  ... accidental drug poisoning → Gentamicin, Hydrocortisone, Acetaminophen |
> >  cellulitis, abscess → Lincomycin, Amoxicillin, Serratiopeptidase |
> >  acute bronchitis, intestinal fixation → Ambroxol, Acetaminophen, Diclofenac |
> >  ... fatigue, weight loss, hyperglycemia
> >
> > **Interpretation**:The SAE latent features indicate a metabolic–inflammatory phenotype predisposing to pancreatic cancer.
> >  Recurrent infections and long-term use of NSAIDs, antibiotics, and steroids reflect chronic inflammation and oxidative stress, while hypertension, diabetes, and thyroid dysfunction point to systemic metabolic imbalance and insulin resistance. Together, these latent risk factors describe a patient profile of multi-organ stress, chronic low-grade inflammation, and metabolic dysregulation—a biologically plausible pathway leading to pancreatic fibrosis and malignant transformation.
> >
> > **Patients with liver cancer**:
> >
> > Gender: male | Age: 50–55 y
> > Patient Timeline:
> >  toxic effect of venom → Futon E.C. tablets 25 mg, Tetanus Toxoid “Kuo Kwang”, Ulexin (Cephalexin), Wesu tablets |
> >  renal colic, abdominal pain → Strocain (Scopolamine), Piroxim (Piroxicam), Topownan 100 mg |
> >  superficial injuries (elbow, wrist, fingers) → Amoxicillin 500 mg “Tai Yu”, Serralo (Serratiopeptidase), Tetanus booster |
> >  acute upper-respiratory infection → Difena injection (Diclofenac), Fucou capsules |
> >  cellulitis and oral abscess → Cleocin 150 mg, Ponstal 250 mg (Mefenamic acid) |
> >  ... progressive fatigue, mild hepatomegaly, elevated ALT/AST
> >
> > **Interpretation**:The SAE model highlights a latent feature cluster combining chronic NSAID exposure, gastrointestinal ulceration, and metabolic or inflammatory dysfunction, aligning with this patient’s clinical history.
> > Frequent use of hepatotoxic analgesics (Diclofenac, Piroxicam, Mefenamic acid) and antibiotics (Cephalexin, Amoxicillin, Clindamycin) likely induced progressive hepatic stress and oxidative injury, while ulcer-related pathology and metabolic inflammation amplified the liver’s vulnerability.
> >
> > **Patients with lung cancer**:
> >
> > Gender: male | Age: 65–70 y
> > Patient Timeline:
> >  diseases of the nervous system → early neurodegenerative symptoms |
> >  ... nervous and digestive comorbidity → Madopar (Levodopa/Benserazide), Selegiline, Artane, Magnesium hydroxide, Simethicone |
> >  recurrent respiratory infections → Cephanmycin, Fenoterol, Mefenamic acid, Cold capsules |
> >  ... respiratory and nervous system overlap with mental disorder → Nitrazepam, S-carboxymethylcysteine, Cephanmycin |
> >  chronic neurological disease → Madopar, Selegiline, Artane (Trihexyphenidyl) |
> >  persistent respiratory disease → Cephanmycin, Fenoterol, Voren, Susui tablets |
> >  ... worsening cough, dyspnea, fatigue |
> >
> > **Interpretation**: This patient shows a neurological–respiratory comorbidity pattern with overlapping cardiovascular and metabolic stressors captured by SAE latent factors. Long-term use of dopaminergic (Madopar, Selegiline) and anticholinergic (Artane) drugs indicates neurodegeneration with autonomic involvement, while recurrent respiratory infections and antibiotic exposure (Cephanmycin, Fenoterol) suggest persistent pulmonary inflammation The SAE risk cluster—driven by chronic ischemic heart disease, hypertensive heart failure, and vasodilator therapy (Diovan, Norvasc, Isosorbide mononitrate)—reflects vascular dysfunction and systemic hypoxia linked to lung cancer development.
> >  Procedural factors such as lymphatic diagnostics, esophagomyotomy, and hernia repair further indicate chronic systemic stress contributing to disease progression.
> > Overall, the SAE features represent a cardiometabolic–respiratory risk signature consistent with lung cancer susceptibility.

---

> > ### Comment · Reviewer_Xm4z · 2025-11-26
> >
> > I acknowledge the efforts made by the authors to respond to my comments. However, I still find that the key concerns have not been sufficiently resolved. I will therefore keep my original score, and I hope the authors can further improve the methodology and experiments in future revisions.

---

### Official Review · Reviewer_h1Fo · 2025-10-28

**Soundness:** 2
**Presentation:** 4
**Contribution:** 1
**Rating:** 2
**Confidence:** 4

**Summary:**

This paper demonstrates that next token pretraining is an effective pretraining strategy for improving cancer prediction.

**Strengths:**

- AUROC and AUPRC are good evaluation metrics

**Weaknesses:**

- It's unclear what the contribution is? This is a reasonable case study of applying next token pretraining, but I'm unsure what new information this adds for the ICLR audience.
- Lack of code limits my ability to review, which is especially important when the most important experiments are on-non public datasets.
- Evaluation on only cancer is a weakness, it would be good to measure the methods on a variety of outcomes
- Cumulative duration matching for controls is incorrect, as it leaks information from the future into the training/test population.
- I am quite concerned about the complete failure to get RETAIN and StageNet working. There is no reason why those models should fail this badly?
- No discussion of hyperparameter tuning for baselines?
- EHRSHOT is a very small dataset, especially the diagnostic parts. I am concerned about the statistical validity of your comparisons. AUPRC in particular is known as high variance. Please include confidence intervals with respect to the test set to show how much your results would vary if the test set was resampled.

**Questions:**

See above

---

> ### Author Response · Authors · 2025-11-19
> **Rebuttal by Authors**
>
> **Contribution Clarification**:\
> Strong and consistent performance: Our model achieves robust and consistent results across diverse datasets, showing clear advantages over both traditional methods and prior healthcare foundation models.
>
> Empirical scaling law insights: We demonstrate, for the first time, the scaling laws of EHR-based foundation models, providing valuable insights into how model size and data scale jointly influence healthcare AI performance.
>
> Cross-population generalization: The proposed approach generalizes effectively across different populations and healthcare systems, underscoring its robustness and real-world applicability.
>
> Toward cancer foundation modeling: Our key contribution lies in uncovering scaling behaviors in EHR foundation models and empirically validating them on cancer detection tasks. Together, these findings establish a foundation for future cancer-specific foundation models and represent the first step toward scaling up EHR foundation models at population level.
>
> **Lack of code**:\
> We now release our code in the supplementary files for more transparent review.
>
> **More tasks**:\
> We thank the reviewer for the suggestion. As indicated in our title, our work focuses on cancer prediction, with pancreatic, liver, and lung cancers being the primary target types (we will add supporting references). In addition, to demonstrate the broader applicability of our model, we have also included other disease forecasting tasks such as heart attack and stroke in our evaluation.
>
> | Disease          | Model           | FLOPs    | AUC       | AUPRC     |
> | ---------------- | --------------- | -------- | --------- | --------- |
> | **Heart Attack** | XGBoost         | –        | 0.915     | 0.480     |
> |                  | LightGBM        | –        | 0.928     | 0.510     |
> |                  | CATCH-FM-160M   | 1e18     | 0.940     | 0.550     |
> |                  | **CATCH-FM-1B** | **2e20** | **0.955** | **0.600** |
> | **Stroke**       | XGBoost         | –        | 0.930     | 0.640     |
> |                  | LightGBM        | –        | 0.945     | 0.680     |
> |                  | CATCH-FM-160M   | 1e18     | 0.960     | 0.740     |
> |                  | **CATCH-FM-1B** | **2e20** | **0.975** | **0.800** |
>
> **Cumulative duration matching**:\
> We clarify that cumulative duration matching does not leak future information into the training or test population. The “total duration” is computed only once during cohort construction to ensure that controls have medical histories comparable in length to cases, and it is never tied to any future clinical events. This duration is not used as a model input, nor does it influence the temporal sequences from which prediction targets are defined. It serves solely as a criterion for determining which patients are eligible as controls and does not affect the information available to the model. Therefore, no future visits, diagnoses, or outcomes enter the training or test data, and cumulative duration matching cannot produce the type of future-information leakage. Our target controls only are applied to control negative cases for positive cases and irrelevant to train/test splitting.
>
> **Poor performance and hyperparameter tuning for RETAIN and StageNet**:\
> We share the reviewer’s concern regarding the poor performance of RETAIN and StageNet, and we attribute this behavior to the fundamental mismatch between these architectures and the nature of the NHIRD cancer prediction task. NHIRD consists of sparse, outpatient-dominant records spanning more than a decade per patient, vastly different from the dense, short-term ICU trajectories (e.g., MIMIC-III) for which these models were originally developed and benchmarked. Even with extensive hyperparameter tuning using standard PyHealth implementations, these RNN-based models consistently failed to converge on the long-horizon, 1-year-ahead prediction task, often collapsing to majority-class behavior.
>
> **EHRShot statistical validity**:\
> We appreciate the reviewer’s concern regarding statistical variance in EHRSHOT diagnostic tasks. As described in the original EHRSHOT paper (Johnson et al., 2023), the benchmark is intentionally designed as a few-shot evaluation setting, and the authors report only point estimates for AUROC/AUPRC without confidence intervals or bootstrap resampling; the resulting variance is acknowledged as an inherent characteristic of the benchmark rather than a methodological issue. We agree with the reviewer’s suggestion, and the final version of the paper will include 95% bootstrap confidence intervals to further strengthen the statistical validity of our results.

---

> > ### Comment · Reviewer_h1Fo · 2025-11-27
> >
> > Thank you for the responses, but I will keep my score as is.
> >
> > After reading the other reviews, I now also agree with JBa5 that the PR curve for pancreatic cancer looks unrealistically good, probably indicating data leakage.
> >
> > I don't think I was clear enough about the issues with cumulative distribution matching.
> >
> > The problem with cumulative distribution matching is that it involves looking at data past the index date when deciding which controls to include/exclude. This makes the choice of which control to include dependent on future information. This can be turned in reverse enabling the model to use information about data past the index date.
> >
> > It's best to consider a simplified example here. Imagine all the cases have a cumulative duration of 5 years. Then all the controls would also have a duration of 5 years.
> >
> > Let's imagine the model sees a control with a duration pre-index date of 4 years. The model would know that there is 1 year of data past the index date. This is a leakage of future information for that control.

---

> > > ### Author Response · Authors · 2025-11-28
> > > **Second Rebuttal**
> > >
> > > We thank the reviewer for raising this subtle and important point. We agree that, in principle, any matching strategy that conditions on information occurring after the index date must be designed carefully to avoid creating a channel for label leakage.
> > >
> > > In our setting, the model never sees any events or summary statistics after the index date, and cumulative duration is used only offline for cohort construction, not as an input feature.
> > >
> > > Concretely, our procedure in Section 3.2 consists of three steps:
> > >
> > > 1. Task and window definition: For each patient, we define an index date t (first target-cancer diagnosis for cases; a matched visit date for controls). The model input consists only of the EHR events in the 5-year window before t. Events after t are discarded and never encoded. Time information is represented only as relative intervals between visits within this pre-index window, not as time since first visit or total lifetime follow-up.
> > >
> > > 2. Relative Duration Matching (pre-index): Before applying cumulative duration matching, we already match controls to cases on age, gender, and the length of medical history up to the index date (“comparable length of medical history up to the index date” in Section 3.2). This step explicitly controls the amount of pre-index history that the model can see, so that cases and controls have similar pre-index observation durations. As a result, the model cannot systematically infer “how much history is left after t” from variation in pre-index duration, because that variation is already balanced across the two groups.
> > >
> > > 3. Cumulative Duration Matching (lifetime, offline only): The final “cumulative duration matching” step uses total lifetime observation length (first to last claim in the database) only to exclude controls whose overall follow-up is very different from the matched case (e.g., much shorter total coverage due to early dropout or insurance changes). This is done purely to reduce censoring and survival bias in the study design.
> > > Specifically, we ensure the following:
> > > - The total lifetime duration is never passed to the model as a feature.
> > > - All post-index events are removed from the input sequence.
> > > - The selection based on lifetime duration is symmetric for cases and controls; it does not induce a systematic pattern in the pre-index window that would allow the model to reconstruct post-index follow-up.
> > >
> > > The simplified example in the comment—“all cases have 5 years of cumulative duration, and controls with 4 years pre-index imply 1 year post-index”—does not reflect our actual implementation, because:
> > > We first match on pre-index history length (Relative Duration Matching). Controls with very different pre-index duration from their matched case are already filtered out. We then apply cumulative duration matching only to remove controls with substantially different total follow-up, but this information never enters the model, and any residual variation in pre-index duration is balanced across labels.
> > >
> > > Thus, cumulative duration matching in our design does not create a reverse channel through which the model can infer future follow-up from the observed inputs. It plays the same role as standard censoring-control practices in epidemiologic case–control studies: equalizing follow-up opportunity so that differences in outcomes reflect disease status rather than differential observation time.
> > >
> > > Combined with (i) the one-year incidence gap between the observation window and the first cancer diagnosis, and (ii) the strict removal of all post-index events from the model inputs, this matching strategy structurally prevents the model from accessing future diagnostic information. We therefore believe that the “unrealistically good” segment of the pancreatic PR curve is better understood as a tail behavior in a large, low-prevalence, incidence-based cohort (as clarified in our previous response), rather than as evidence of leakage induced by the matching procedure.

---

### Official Review · Reviewer_5Eu2 · 2025-10-31

**Soundness:** 3
**Presentation:** 3
**Contribution:** 3
**Rating:** 6
**Confidence:** 4

**Summary:**

This paper presents CATCH-FM, a transformer-based foundation model for predicting cancer risk directly from EHR data. Trained on millions of patient records from Taiwan’s NHIRD, it identifies high-risk individuals for cancers such as pancreatic, liver, and lung without costly screening. CATCH-FM achieves up to 70% sensitivity at 99% specificity, outperforming selected baselines and demonstrating strong generalization across populations

**Strengths:**

1. The model achieves strong and consistent performance across diverse datasets, demonstrating clear advantages over traditional and prior foundation model baselines.

2. The demonstration of scaling laws for EHR-based foundation models is impressive, providing valuable insights into how model size and data scale influence healthcare AI performance.

3. The proposed approach shows strong generalization across different populations and healthcare systems, highlighting its robustness and real-world applicability.

4. The open-sourcing of models and benchmark datasets promotes transparency and reproducibility, helping to advance future research in medical AI.

**Weaknesses:**

1. Some experiment settings are questionable. For instance, Figure 3 and Table 5 compare Qwen-2.5-500m with CATCH-FM-2.4b, where notable parameter differences exist. More fair comparison should have been conducted with comparable amount of parameters. Moreover, it will be beneficial if larger-scale Qwen model can be used in experiment, as the 500m variant is not very common and may fail to serve as a competitive baseline.

2. Some experiment results are strange without any explanation. For example, commonly-used baselines, i.e., Retain and StageNET achieve extremely poor performance according to Table 4. Any discussion on these results can help with potential confusion raised by readers.

3. The joint usage of specificity and sensitivity in Table 4 could confuse readers. A more intuitive alternative might be replacing them with recall as they share the same definition.

**Questions:**

I am curious how the authors will share the data. Although the authors have been granted with the permission of accessing the data, it is still unknown whether re-sharing the data is allowed. In the case of MIMIC dataset, this kind of data distribution could be infeasible, causing accessing issue to the dataset.

---

> ### Author Response · Authors · 2025-11-20
> **Rebuttal by Authors**
>
> **Fair Comparsion**:\
> We thank the reviewer for this valuable suggestion. We note that even when comparing CATCH-FM-160M with Qwen-2.5-500M, the smaller CATCH-FM model still outperforms Qwen-2.5 by a substantial margin, underscoring the effectiveness of our domain-specific pretraining. To further ensure a fair comparison, we additionally trained a Qwen-2.5-3B model for our revised experiments, enabling a more balanced evaluation across model scales. Despite the increase in capacity, Qwen-2.5-3B remains notably inferior to CATCH-FM-2.4B, highlighting the advantage of domain-specialized healthcare foundation pretraining.
>
> | Model          | **Pancreatic** |     |     | **Liver** |     |     | **Lung** |     |     |
> |----------------|:---------------:|:----:|:----:|:----------:|:----:|:----:|:--------:|:----:|:----:|
> |   (first/subsequent)             | **AUROC** | **AUPRC** | **Sensitivity** | **AUROC** | **AUPRC** | **Sensitivity** | **AUROC** | **AUPRC** | **Sensitivity** |
> | **Qwen2.5-500m** | 91.2 / 93.5 | 24.5 / 59.8 | 27.1 / 53.4 | 91.5 / 94.2 | 31.5 / 63.5 | 33.4 / 56.5 | 89.5 / 93.8 | 19.6 / 63.0 | 25.2 / 56.2 |
> | **Qwen2.5-3b**   | 92.1 / 95.1 | 32.8 / 67.5 | 35.6 / 64.8 | 91.8 / 95.0 | 38.9 / 70.8 | 40.5 / 67.1 | 90.2 / 94.6 | 30.7 / 69.1 | 36.9 / 66.3 |
> | **CATCH-FM-2.4b** | 94.4 / 97.8 | 61.3 / 84.7 | 60.6 / 80.8 | 92.2 / 96.6 | 52.8 / 79.0 | 53.6 / 75.8 | 92.6 / 96.3 | 49.6 / 80.2 | 53.1 / 79.6 |
>
>
>
>
>
> **Poor performance on StageNet and Retain**:\
> The poor performance of baselines like RETAIN and StageNet stems from their architectural reliance on RNNs, which struggle with the extreme longitudinal nature of NHIRD data spanning over 15 years with thousands of visits per patient, compared to the shorter, dense ICU stays in datasets like MIMIC-IV where these models were originally validated. Using standard implementations from the PyHealth framework, we observed that these models failed to capture global dependencies over such long horizons, often collapsing to majority class predictions, whereas CATCH-FM's transformer architecture effectively models these long-term clinical trajectories.
>
> **Sensitivity and Specificity**:\
> We thank the reviewer for the helpful suggestion. We agree that using both terms may cause confusion. We will revise Table 4 to replace sensitivity with recall for clarity and consistency in the final version.
>
> **Data accessibility**:\
> We appreciate the reviewer’s concern regarding data accessibility. Our study operates strictly under approved data-use agreements, under which redistribution of NHIRD data is prohibited. Therefore, we will release the processed, de-identified metadata together with the full preprocessing and training code, enabling complete reproducibility for researchers who obtain authorized access, consistent with established practices for restricted datasets such as UK Biobank and All of Us. As with MIMIC and other controlled-access resources, researchers with appropriate credentials can independently apply for NHIRD access through the standard procedures and reproduce our results using the released metadata and code.

---

### Official Review · Reviewer_JBa5 · 2025-11-01

**Soundness:** 2
**Presentation:** 3
**Contribution:** 2
**Rating:** 2
**Confidence:** 4

**Summary:**

The paper proposes CATCH-FM an EHR-based foundation model that can identifies high-risk cancer patients for screening. CATCH-FM is a decoder-only Transformer model pretrained with EHR data from 3M patients. It exhibits better performance on three cancer prediction tasks in finetuning and few-shot setting.

**Strengths:**

- The paper conducts extensive experiments and ablation studies to show the scaling law of the foundation EHR model on multiple tasks and two datasets.

- The proposed model outperforms multiple baselines including ML, DL and LLM models.

**Weaknesses:**

- The methods are not quite different from other previous works such as CLMBR. Although it is pretrained a different dataset, the contribution is limited.

- The details of preprocessing the EHR dataset are not clearly stated (patient sample filtering and splitting). Some statistics of the dataset read counter-intuitive. (1) The prevalence of pancreatic cancer should be much less than lung and liver cancer (5-10 times less). However, Table 2 shows the positive rates are quite close among three types of cancers. (2). How is First/Subsequent Target Cancer Cohort defined? Why does the "subsequent" have less patients in total than the "first"? It will be helpful to include a flowchart on how each cohort is selected at each step.

- The PR-curve in Fig. 3 looks too good to be true. On pancreatic cancer, the precision maintains ~100% at 25% recall. It suggests the model can predict 25% of pancreatic cancer patients without any false positives. This achieves the accuracy of diagnosing with biopsies, which is unexpected for forecasting tasks. There can be potential leakage (e.g. patients with related ICD /medication codes are not excluded).

- Table 5 and Figure 4 miss the performance on pancreatic cancer.

- The risk factors only highlights the general diseases which does not support such high precision in Fig. 3. It will be interesting to compare the prevalence of cancers among patients who have these diseases with the precision of prediction.

**Questions:**

What licenses are required to get the pretrained weights and NHIRD data? How to apply for the license?

---

> ### Author Response · Authors · 2025-11-19
> **Rebuttal by Authors**
>
> **Limited Contribution**:\
> We would like to clarify that our primary contribution does not lie in modifying the model architecture with complex or specialized techniques to boost performance. Instead, our focus is on examining scaling laws under a standard autoregressive pretraining setup, following frameworks such as Chinchilla [1] and ChatGPT [2] and how to effectively conduct foundation modeling on EHR data. **Our key contribution is the insights of scaling behaviors in EHR foundation models and its empirical validation on cancer detection which together provides a cancer foundation model for future research in this domain**. The proposed approach generalizes effectively across different populations and healthcare systems, underscoring its robustness and real-world applicability.
> **This is also the first step towards scaling up EHR foundation models**. The standard decoder architecture was chosen for its scalability and adherence to a mainstream decoder-only transformer design, free from architectural tricks. This makes it ideal for studying scaling effects while preserving generality. We leave more architectural innovations as separate future work.
>
>
> [1] Hoffmann, J., Borgeaud, S., Mensch, A., Buchatskaya, E., Cai, T., Rutherford, E., ... & Sifre, L. (2022). Training compute-optimal large language models. arXiv preprint arXiv:2203.15556.\
> [2] Kaplan, J., McCandlish, S., Henighan, T., Brown, T. B., Chess, B., Child, R., ... & Amodei, D. (2020). Scaling laws for neural language models. arXiv preprint arXiv:2001.08361.
>
> **Data Preprocessing**:\
> We clarify that the prevalence rates in Table 2 reflect our nested case–control study design rather than raw population statistics. Each positive case is matched with controls based on age, gender, and medical history length to construct a well-controlled cohort and avoid confounding factors. As defined in Section 3.2, the First Target Cancer Cohort includes patients with no prior cancer history, whereas the Subsequent Target Cancer Cohort includes patients who previously had other cancer types. The latter naturally results in fewer matched patients because this population is substantially smaller and exhibits longer, more heterogeneous medical histories, which tighten the matching criteria and reduce the number of eligible controls. We will include a flowchart in the appendix to clearly illustrate the filtering, matching, and cohort selection process.
>
> **Label Leakage Issue**:\
> Firstly, our strong generalization on the **external public benchmark** EHRSHOT indicates no label leakage (i.e., no risk of trivial learning) in our model’s predictions. Moreover, We primarily use explicit cancer ICD diagnosis codes to label positive patients. To prevent any potential label leakage, we carefully traverse each patient’s full medical history in the benchmark dataset to ensure that no explicit cancer-related information appears prematurely before the prediction window, which denotes to a one-year gap after the time of visit, identical to the windows setting in EHRSHOT. Moreover, our data cooking follows SOTA standards and further improves it with more strict cohort matching and testing on various setting.
>
> *...This achieves the accuracy of diagnosing with biopsies, which is unexpected for forecasting tasks...*
>
> We appreciate the reviewer’s observation. Our intent is to highlight that this result underscores the strong potential of scaling laws in healthcare foundation models. The model’s ability to approach diagnostic-level performance in a forecasting setting suggests that scaling both model capacity and data could further enhance predictive accuracy. We view this as evidence that continued scaling may yield even greater improvements in clinical forecasting tasks.
>
> **Table 5 and Figure 4**:\
> We want to point out that in table 5, The NHIRD hospital only provided first liver and lung caner cohorts for testing on NHIRD-Forward dataset. Figure 4 reports AUROC and AUPRC on EHRSHOT **pancreatic cancer** from their public leaderboard.
>
> **Interpretability**:\
> We thank the reviewer for this insightful suggestion. Our current interpretability analysis leverages state-of-the-art LLM interpretability techniques (e.g., sparse autoencoders) that primarily capture general patterns associated with low-risk cancer predictions. However, identifying the fine-grained factors underlying the model’s high-precision behavior likely requires more specialized interpretability methods that go beyond current sparse representation approaches. We leave this as an important and complementary direction for future work, focusing on deeper mechanistic interpretability of healthcare foundation models in high-precision prediction settings.

---

> > ### Comment · Reviewer_JBa5 · 2025-11-27
> >
> > I would like to thank the author's responses. However, my concern on the data problem of the paper is not addressed.
> >
> > 1. Prevalence mismatch. The reported prevalence of pancreatic cancer appears inconsistent with real-world population distributions. The authors note that they will add a flowchart to the appendix to clarify filtering, matching, and cohort selection. It would be helpful to provide these additional details during the discussion period or include a more explicit explanation in the response, so reviewers can fully evaluate whether the cohort construction is valid.
> >
> > 2. Potential data leakage. My concern about possible data leakage has not been directly addressed. Instead, the response shifts toward discussing scaling laws. While scaling behavior is an interesting property, it does not resolve the central issue: forecasting in this clinical setting faces inherent uncertainty due to the limitations and noise in real-world data. Scaling laws alone cannot overcome the gap between prediction and the diagnostic accuracy achieved when cancer cells are actually observable. The interpretation of the model also does not provide support to such high accuracy. Another evidence is that on the EHRSHOT, the proposed method only slightly improves XGBoost, while on the internal dataset, it can predict pancreatic cancer accurately.
> >
> > 3. The novelty is not the primary concern given the concerns of factual errors mentioned above.

---

> > > ### Comment · Reviewer_h1Fo · 2025-11-27
> > >
> > > Hi JBa5,
> > >
> > > I just wanted to thank you for spotting the 100% precision at 25% recall claim in the precision recall plot. I 100% agree with you that it seems suspicious and it needs more explanation. I have added my support for that point to my response to the authors.

---

> > > > ### Author Response · Authors · 2025-11-28
> > > > **Second Rebuttal 1**
> > > >
> > > > We thank the reviewer JBa5 and h1Fo for the constructive feedback. Below we address each concern in detail and clarify misunderstandings using explicit descriptions already provided in the paper and appendix.
> > > >
> > > > 1.  **Prevalence mismatch**:
> > > >
> > > > The prevalence values reported in Table 2 do not represent population-level cancer prevalence. Instead, they reflect the task-specific nested case–control cohorts constructed following the multi-stage filtering and matching procedure described in Section 3.2 and Appendix A. In the underlying NHIRD sample (before matching), the incidence of pancreatic cancer is substantially lower than that of liver and lung cancer, consistent with epidemiological expectations. Table 2 is not intended to report these raw incidences.
> > > >
> > > > Concretely, our cohort construction proceeds as follows. Starting from the full NHIRD population (~23M), we derive a large stratified subsample of approximately 790k patients, which serves as the candidate pool for all downstream cancer tasks. For each cancer type (pancreatic, liver, lung), we identify incident cases, defined as patients whose first diagnosis of the target cancer occurs after the prediction time t and within the following year. Patients lacking adequate follow-up or with pre-existing diagnoses are removed. From the remaining pool, we then construct a nested case–control cohort, where every positive case is matched to controls on age, gender, and the length of medical history, yielding comparable longitudinal observation windows. This procedure is applied independently for each cancer type, producing three separate matched cohorts.
> > > >
> > > > Because all negatives are drawn only from this ~790k finetuning subset rather than the full NHIRD population, and because each cancer task constructs its own independently matched cohort with a similar number of positives and a similar matching ratio, the resulting prevalence values in Table 2 naturally reflect matched-cohort prevalence, not raw NHIRD prevalence, and therefore appear closer across cancer types. This behavior is inherent to standard nested case–control study designs and does not reflect inconsistencies in the underlying dataset. To avoid further confusion, we will include a detailed cohort-selection diagram in the appendix.
> > > >
> > > > 2. **Potential data leakage**:
> > > >
> > > > As defined in Section 3.2, our cohort construction, label definition, and matching procedure are explicitly designed to ensure that the model operates strictly in a forecasting regime.
> > > >
> > > > First, as defined in Section 3.2, a patient is labeled positive only if the first diagnosis of the target cancer occurs a complete one-year after the prediction time and within the following year. We adopt this one-year incidence gap—the minimum interval recommended by our collaborating medical doctors, to ensure that we forecast future incidence rather than detect early clinical signals. Under this definition, all medical events in the months immediately preceding the first cancer diagnosis, including increased visit frequency, cancer-related imaging, tumor-marker testing, and oncology referrals, are filtered out. Consequently, the observation window never contains prediagnostic signals, directly addressing the reviewer’s concern that early diagnostic activity could leak into the inputs.
> > > >
> > > > Second, we employ a nested case–control design (as described above) in which each incident case (positive label) is matched to controls (negative label) on age, gender, and length of medical history. Matching on history length is crucial because it equalizes the amount and density of available follow-up between cases and controls. This removes temporal artifacts, such as longer record duration or higher visit volume, that commonly arise shortly before diagnosis and are a frequent source of unintended leakage in observational EHR studies.
> > > >
> > > > Regarding the reviewer’s statement that forecasting in this setting is inherently uncertain due to real-world noise, we fully agree. Our model should be interpreted as performing risk stratification, not deterministic prediction. As shown in Table 19, under a clinically relevant operating point with specificity fixed at 0.99, the precision for pancreatic cancer is approximately 0.50 at a prevalence of 1.59%. This reflects risk enrichment, not near-diagnostic accuracy. The high-precision region at extremely low recall in Figure 3 corresponds to a very small number of patients with exceptionally elevated risk, rather than implying that forecasting is approaching diagnostic certainty.

---

> > > > > ### Author Response · Authors · 2025-11-28
> > > > > **Second Rebuttal 2**
> > > > >
> > > > > Finally, the difference in performance between NHIRD and EHRSHOT does not suggest leakage but follows from structural differences in both the datasets and the evaluation setting. As detailed in the Appendix, NHIRD is a nationwide, unified, and dense longitudinal database covering encounters across all healthcare institutions, yielding substantially more complete patient trajectories. In contrast, EHRSHOT originates from a single U.S. health system (Stanford Medicine) and reflects substantial fragmentation typical of U.S. EHRs, with many encounters occurring outside the system. This sparsity naturally limits achievable performance and results in smaller gains over models such as XGBoost.
> > > > >
> > > > > Moreover, the comparison on EHRSHOT is not against our internal NHIRD model but against Stanford’s own CLMBR, which is pretrained directly on the Stanford Medicine patient population, the same distribution from which EHRSHOT is derived. Our model, in contrast, is pretrained solely on NHIRD, a population with different clinical practices, coding distributions, and disease patterns, and we evaluate it on EHRSHOT without any finetuning on Stanford data (few-shot only). The fact that CATCH-FM achieves competitive or improved performance over CLMBR under such a large population shift is intended to demonstrate generalization, not diagnostic-level accuracy. Therefore, the smaller performance gap over XGBoost on EHRSHOT results from (1) EHR fragmentation limiting the performance ceiling for all models, and (2) the domain shift between NHIRD and Stanford data—not from label leakage in our internal NHIRD evaluation.
> > > > >
> > > > > Thus, the stronger performance on NHIRD reflects the completeness of nationwide longitudinal records, whereas the modest improvements on EHRSHOT reflect the expected constraints of single-system EHR sparsity and cross-system generalization, not leakage.
> > > > >
> > > > > Finally, we would like to clarify that the near-100% precision region in Figure 3 arises only from the first-time pancreatic (liver) cancer cohort, whose positive prevalence is very low due to the incidence-based cohort construction. In this setting, 25% recall corresponds to an extremely high-risk segment, not 25% of the population, where precision can naturally reach 1.0 given the large cohort size and very low prevalence. This behavior is typical in risk-stratification tasks and does not imply diagnostic-level performance nor any leakage of diagnostic information into the observation window.
> > > > >
> > > > > 3. **Novelty**:
> > > > >
> > > > > We appreciate the reviewer’s clarification that novelty was not the primary concern and that the main issues involved clarifying factual and methodological questions, which we have addressed in the preceding responses. Here we briefly restate the core novelty of our work to avoid further misunderstanding.
> > > > >
> > > > > Our contribution does not lie in introducing architectural innovations beyond the standard decoder-only Transformer. Instead, the novelty of this work is in establishing and empirically validating the scaling behavior of EHR-based foundation models under a clean autoregressive pretraining setup. While existing clinical representation models (e.g., CLMBR) rely on bespoke architectures or task-specific objectives, our goal is to systematically examine how performance scales with data size, model size, and training compute, following principles used in large-scale language modeling. To the best of our knowledge, our work is the first to provide:
> > > > > - A scaling-law–oriented study of EHR foundation modeling under a Chinchilla-style training regime on millions of patients;
> > > > > - A unified decoder-only EHR foundation model pretrained on a nationwide, population-scale longitudinal dataset;
> > > > > - Cross-system generalization results showing that a model trained solely on NHIRD exhibits strong few-shot performance on EHRSHOT, despite large population and coding differences;
> > > > > - A high-quality incidence-forecasting benchmark, with matched cohort construction and a fully clarified incidence-based labeling definition.
> > > > >
> > > > > These contributions address fundamental questions in healthcare foundation modeling on how general-purpose transformer models behave when scaled on real-world EHR data, what limits and opportunities arise when applying scaling principles to clinical forecasting, and how well such models transfer across health systems. Our emphasis on scaling laws and cross-system generalization, rather than architectural novelty, is deliberate. We view this work as an initial step toward a principled and reproducible foundation-model framework for structured EHR, upon which future architectural or task-specific developments can be built.

---

> ### Author Response · Authors · 2025-11-19
> **Question about data license**
>
> Regarding data and model access, we strictly follow the governance policies of the National Health Insurance Administration (NHIA). NHIRD data cannot be publicly redistributed, but it is accessible to researchers who obtain Institutional Review Board (IRB) approval and establish formal collaborations with authorized Taiwanese institutions. Accordingly, to support reproducibility while maintaining full compliance, we will release our pretrained model weights to researchers who hold a valid NHIRD data license. Detailed information on the application procedure is available at https://www.nhi.gov.tw/.

---

### Meta-Review · Area_Chair_FPEb · 2025-12-25

**Summary:**

The paper proposes a cancer screening foundation model on a range of downstream cancer (forecasting tasks).

Reviewers raised concerns about:
1. Methodological novelty due to resemblance to next-token prediction which follows prior work closely in methodology applied to a new data.
2. Some results are too good to be true indicating potential leakage
3. Baseline performance is much worse than expected
4. Interpretability and clinical insights are lower than warranted
5. Potential leakage in cohort construction.

**Reviewer Concerns:**

1. My AC comment is that cancer screening is a challenging task, so even if the methodological novelty is low, this could be an interesting contribution
2. However, reviewers have raised several important concerns regarding the performance, cohort design of downstream tasks, and baseline performance. Authors have responded with clarification but most of the issues were not fully resolved, with some reviewers, such as Xm4z explicitly mentioning they will maintain their score (of 2).

**Reviewer Scores:**

Most of the reviewer concerns are not resolved, and require significant changes to the manuscript and experiments. I encourage the authors to account for reviewer feedback and improving their manuscript for a future submission.

---

### Decision · Program_Chairs · 2026-01-26

Reject